cellular biology/nanotechnology

magnetite nanoparticles, magnetic hyperthermia, endocytosis, cell uptake

**Authors for correspondence:**
Keya Mao
e-mail: maokeya@sina.com
Peifu Tang
e-mail: pftang301@126.com

†These authors contributed equally to the study.

# The cell uptake properties and hyperthermia performance of $Zn_{0.5}Fe_{2.5}O_4/SiO_2$ nanoparticles as magnetic hyperthermia agents

Runsheng Wang[1,2,†], Jianheng Liu[1,†], Yihao Liu[1], Rui Zhong[1], Xiang Yu[3], Qingzu Liu[1], Li Zhang[3], Chenhui Lv[3], Keya Mao[1] and Peifu Tang[1]

[1]Medical School of Chinese PLA, Beijing 100853, People's Republic of China
[2]Department of Orthopedics, The Third Affiliated Hospital of Guangxi Traditional Chinese Medicine University, Liuzhou, Guangxi Zhuang Autonomous Region 545001, People's Republic of China
[3]Department of Physics, Capital Normal University, Beijing 100048, People's Republic of China

RW, 0000-0001-9941-5027

$Zn_{0.5}Fe_{2.5}O_4$ nanoparticles (NPs) of 22 nm are synthesized by a one-pot approach and coated with silica for magnetic hyperthermia agents. The NPs exhibit superparamagnetic characteristics, high-specific absorption rate (SAR) ($1083\ wg^{-1}$, $f = 430\ kHz$, $H = 27\ kAm^{-1}$), large saturation magnetization ($M_s = 85\ emu\ g^{-1}$), excellent colloidal stability and low cytotoxicity. The cell uptake properties have been investigated by Prussian blue staining, transmission electron microscopy and the inductively coupled plasma-mass spectrometer, which resulted in time-dependent and concentration-dependent internalization. The internalization appeared between 0.5 and 2 h, the NPs were mainly located in the lysosomes and kept in good dispersion after incubation with human osteosarcoma MG-63 cells. Then, the relationship between cell uptake and magnetic hyperthermia performance was studied. Our results show that the hyperthermia efficiency was related to the amount of internalized NPs in the tumour cells, which was dependent on the concentration and incubation time. Interestingly, the NPs could still induce tumour cells to apoptosis/necrosis when extracellular NPs were rinsed, but the cell kill efficiency was lower than that of any rinse group, which indicated that local temperature rise was the main factor that induced tumour cells to death. Our findings suggest that this high

SAR and biocompatible silica-coated $Zn_{0.5}Fe_2O_4$ NPs could serve as new agents for magnetic hyperthermia.

# 1. Introduction

Magnetic nanoparticles (MNPs) have attracted tremendous attention for their diverse biomedical applications including drug delivery [1], imaging [2], tumour targeting with various chemotherapeutic drugs [3], cell separation [4], gene therapy [5] and magnetic hyperthermia [6]. Hyperthermia is used as an adjunctive treatment in tumour therapy, the mechanism of hyperthermia is that tumour cells are much more sensitive to heat shock than normal healthy cells when exposed to the temperature range of 43–45°C, their proliferation and metabolic activity are inhibited which can lead to either apoptosis or necrosis [7]. Intracellular magnetic hyperthermia was proposed by Gordon in 1979 [8], which had more advantages than conventional hyperthermia (such as microwaves, radiofrequency and high-intensity-focused ultrasound), resulting in heating the tumour cells from 'inside-out' which can prevent unexpected damage to the surrounding normal tissues [9]. Current studies have investigated the excellent hyperthermia performance of MNPs when internalized into tumour cells [10]. Also, types of core–shell iron oxide nanoparticles (IONP) have been designed to enhance the target to organelles of tumour cells, which can improve hyperthermia performance [11,12]. Bi-functional even multi-functional nanoparticles (NPs) have been designed to trigger tumour cells to apoptosis or necrosis by alternating magnetic fields or lasers [13,14]. Although pulsed magnetic fields can improve the transport of IONP through the cell membrane, which represents increasing the amount of cell internalization [15], some types of NPs even can motivate apoptosis or necrosis of cancer cells without raising the temperature in the special magnetic fields, such as dynamic magnetic fields (DMF) [16,17]. Although the structures of MNPs are different, the mechanisms of heating emission are the same. Two main magnetic relaxation mechanisms determine the thermal efficiency of NPs, i.e. the magnetic process (Néel relaxation) by which the magnetic moments fluctuate without NP rotation and the mechanical process (Brownian relaxation) by which the NPs rotate with their magnetic moment fixed at a given crystal axis [18].

However, the mobility of NPs, undergoing Brownian relaxation, will be inhibited when NPs internalize into tumour cells, then the heating efficiency decreases because the NPs can only undergo Néel relaxation [19]. This phenomenon has been proved in *in vitro* [20] and *in vivo* [21] studies. Meanwhile, for the safety of patients, the product of magnitude and frequency of alternating magnetic fields must be less than $5 \times 10^9 \ Am^{-1} \ s^{-1}$ [22]. Consequently, hyperthermia performance can be improved by increasing NP concentration, but the long-term toxicity of NPs is still unclear; previous studies have demonstrated that the breakdown and clearance of heavy metals included in NPs are quite slow [23]. In summary, NPs with a high specific absorption ratio (SAR) are needed to reduce the amount of particles delivered to the body regarding the as-yet incompletely known toxicity. The higher SAR can partly compensate for the lower concentration of particles in the tumour [19], while coated with silica it can reduce the cytotoxicity and prevent NPs from aggregation and oxidation [24].

Cell uptake is also called internalization or endocytosis, which is a cellular process in which substances penetrate into the cell. The material to be internalized is surrounded by an area of the plasma membrane, which then buds off inside the cell to form a vesicle containing the ingested material. Endocytosis includes pinocytosis (cell drinking) and phagocytosis (cell eating). The interactions between NPs and cells are critical when NPs are considered as biomedical applications [25]. Cell uptake properties are the crucial factors that may affect the magnetic hyperthermia efficiency. The size and surface modifications of NPs lead to different responses concerning cell interaction [26]. Riccardo [20] *et al.* revealed that the fall in heating efficiency correlated with a complete inhibition of Brownian relaxation in cellular conditions. Soukup *et al.* [19] verified the block of Brownian relaxation through the AC susceptibility signal from internalized NPs in live cells, which was consistent with the measurements of immobilized NP suspensions. However, Cabrera *et al.* [27] suggested that the enhancement of intracellular IONP clustering mainly led to the decrease in heating efficiency rather than intracellular IONP immobilization. Hence, a detailed understanding of cell uptake properties is crucial when NPs are considered as hyperthermia meditators.

Here we synthesized high heating efficiency and low cytotoxicity silica-coated zinc doping IONP; the cell uptake properties and intracellular hyperthermia performance were investigated. Our investigation demonstrated that $Zn_{0.5}Fe_{2.5}O_4/SiO_2$ NPs exposed to human osteosarcoma MG-63 cells resulted in

time-dependent and concentration-dependent internalization. The internalization appeared between 0.5 and 2 h after incubation. The intracellular hyperthermia efficiency was related to the amount of internalized NPs in tumour cells. It is worth mentioning that the tumour cells could be killed when the extracellular NPs were rinsed. Our finding suggests that such biocompatible silica-coated $Zn_{0.5}Fe_{2.5}O_4$ NPs with high SAR could serve as new agents for magnetic hyperthermia.

# 2. Material and methods

## 2.1. Nanoparticle synthesis and characterization

The synthesis methods of $Zn_{0.5}Fe_{2.5}O_4$ NPs and silica coating has been reported before [28].

### 2.1.1. Materials

Zinc (II) acetylacetonate, iron (III) acetylacetonate and sodium oleate were purchased from Alfa Aesar. Oleate (90%), oleic acid (90%), benzyl ether (98%), lgepal CO-520, cyclohexane, tetraethyl orthosilicate (TEOS) and ammonium hydroxide were purchased from Sigma-Aldrich. Ethanol and hexane were purchased from Beijing Chemical Co., Ltd. (China).

### 2.1.2. Synthesis of $Zn_{0.5}Fe_{2.5}O_4$ nanoparticles

Zinc (II) acetylacetonate (0.5 mmol), iron (III) acetylacetonate (2.5 mmol), oleic acid (4 ml) and sodium oleate (2 mmol) were mixed with benzyl ether (20 ml) under an argon flow. The solution was stirred by a magnetic stirring apparatus under the argon atmosphere and heated to 393 K for 1 h. The mixture was further heated to reflux temperature ($\approx$573 K) and was maintained for 1 h. After cooling down to room temperature, the NPs were collected by centrifugation. The size of magnetic NPs rested with the heating rate from 393 to 573 K.

### 2.1.3. Silica coating of nanoparticles

The hydrophobic NPs were turned into hydrophilic solutions by coating silica shells via a reverse microemulsion method. Cyclohexane (10 ml) and lgepal CO-520 (0.575 ml) were mixed by an ultrasonic agitator for 10 min, then magnetic NPs (10 mg) in cyclohexane (1 ml) were added and stirred for 30 min. Ammonium hydroxide (0.075 ml, 28–30%) was then added and followed by 0.05 ml TEOS. The solution was sealed and stirred at room temperature for 24 h. Then ethanol and hexane were added and the $Zn_{0.5}Fe_{2.5}O/SiO_2$ core/shell NPs were collected by centrifugation. Finally, the core/shell NPs were washed twice in ethanol and deionized water and collected by centrifugation and redispersed in deionized water.

### 2.1.4. Characterization

The morphology and size of NPs were characterized by a high-resolution Hitachi H7650 (120 kV) transmission electron microscope (TEM). The structural formations of NPs were determined by X-ray diffraction (XRD) (D8 ADVANCE, Bruker Corp., DE). A vibrating sample magnetometer (VSM) (3473-70, GMW Corp., NZ) was used to measure the magnetic hysteresis loops. To investigate the hyperthermia performance of $Zn_{0.5}Fe_{2.5}O_4/SiO_2$, the SAR was characterized by an alternating magnetic field (AMF) (HYPER5, MSI Corp., USA) with a frequency of 430 kHz. A fibre-optic probe (Neoptix Corp., CA) was applied to investigate the temperature of the NP solution every 1 s after turning on the magnetic field.

## 2.2. Cell culture

Mouse embryonic fibroblast (MEF) cells and human osteosarcoma MG-63 cells were all purchased from the American Type Culture Collection (ATTC, USA). Cells were all cultured in a DMEM/HIGH GLUCOSE (Hyclone, USA) medium supplemented with 10% fetal bovine serum (FBS) and antibiotics (100 µg ml$^{-1}$ penicillin and 100 µg ml$^{-1}$ streptomycin) in a humidified atmosphere of 5% $CO_2$ at 37°C.

## 2.3. Cytotoxicity

Cell Counting Kit-8 (CCK-8, Dojindo, Japan) assay was used to evaluate the cytotoxicity of $Zn_{0.5}Fe_{2.5}O_4/SiO_2$ to human osteosarcoma MG-63 and MEF cells. For the CCK-8 assay, the MEF and human osteosarcoma MG-63 cells were incubated in 96-well plates at a density of $1 \times 10^4$ per well and grown overnight ($n = 5$ per group) and then co-incubated with various concentrations (0, 25, 50, 100, 200, 400, 800, 1000 µg ml$^{-1}$) of $Zn_{0.5}Fe_{2.5}O_4/SiO_2$ at 37°C for 24 and 48 h. Following this incubation, each well was washed with phosphate-buffered saline (PBS) three times. Then the cells were incubated in media with 10% CCK-8 for 1 h. After this incubation period, the solution was transferred to a blank 96-well plate and the NPs were fixed with a permanent magnet under the bottom of the plate to avoid the affection of NPs in the process of optical density (OD) measurement. The absorbance was measured at 450 nm with a multimode plate reader (TECAN InfiniteM200 PRO, Switzerland). Cell viability was expressed as the percentage of viable cells compared with controls (cells treated with a culture medium).

## 2.4. Cell uptake

### 2.4.1. Prussian blue staining

The cellular uptake of $Zn_{0.5}Fe_{2.5}O_4/SiO_2$ NPs to human osteosarcoma MG-63 cells was measured by Prussian blue staining (Solarbio, Beijing, China). Various concentrations (6.25, 12.5, 25, 50, 100, 200 µg ml$^{-1}$) of $Zn_{0.5}Fe_{2.5}O_4/SiO_2$ NPs were incubated with human osteosarcoma MG-63 cells in 24-well plates at a density of $10^5$ cells per well ($n = 4$ per group). After 24 h incubation, the cell layers were washed with PBS three times and fixed with 4% paraformaldehyde for 10 min in 4°C. To stain the intracellular iron, the Prussian blue solution mixed with a 2% hydrochloric acid aqueous solution and 2% potassium ferrocyanide (II) trihydrate was incubated with the fixed cells for 30 min in 37°C. Then, the fixed cells were washed with ultrapure water three times and counterstained with nuclear eosin for 20 s. After washing with ultrapure water, the cells were placed on a microscope for cellular uptake observation. $Zn_{0.5}Fe_{2.5}O_4/SiO_2$ NPs (200 µg ml$^{-1}$) were incubated with human osteosarcoma MG-63 at different times (0.5, 2, 4, 6, 8, 10, 12 h) in 24-well plates at a density of $10^5$ cells per well ($n = 4$ per group). After incubation, the intracellular iron was measured by standard Prussian blue staining, as described previously.

### 2.4.2. Transmission electron microscopy

TEM images were recorded with an FEI Tecnai G2 spirit microscope (Thermo Fisher Scientific, USA) operating at an accelerating voltage of 120 kV. For the characterization of the doped cells, after incubation with $Zn_{0.5}Fe_{2.5}O_4/SiO_2$ NPs, the human osteosarcoma MG-63 cells ($1 \times 10^6$) were washed three times with PBS and fixed with 1.5% glutaraldehyde in PBS at 4°C for 30 min. The fixed cells were washed three times with PBS and 1% osmium tetroxide in PBS was added for 1 h at room temperature. After another three washing steps in PBS, the cells were dehydrated with 30, 50, 75, 85, 95 and 100% (three times) absolute ethanol. Thereafter, the cells were infiltrated with Epon resin (two steps: 50 and 66% for resin in absolute ethanol, 30 min each) and embedded in 100% resin at 60°C for 2 days. Ultrathin sections (70 nm thick) were cut on an Ultramicrotome (Leika), stained with lead citrate and observed by TEM.

### 2.4.3. Inductively coupled plasma–mass spectrometry

$Zn_{0.5}Fe_{2.5}O_4/SiO_2$ NPs at a concentration of 200 µg ml$^{-1}$ were incubated with human osteosarcoma MG-63 cells ($1 \times 10^6$) for different times (0.5, 2, 4, 6, 8, 10, 12 h). Then, the extracellular NPs were rinsed with PBS three times, the cells were detached with 2.5% trypsin-EDTA solution and collected by centrifugation (1000 rpm × 5 min). The samples were digested in concentrated 3 : 1 HCl/HNO3 (v/v) solutions, the amount of internalization iron was determined by an elemental analysis using an inductively coupled plasma–mass spectrometer (ICP-MS) (Thermo Fisher Scientific, USA). All time points have been acquired in triplicate.

## 2.5. Intracellular hyperthermia

### 2.5.1. Cytotoxicity of intracellular hyperthermia

Human osteosarcoma MG-63 cells were plated in 35 mm inner diameter culture dishes at 50% confluence ($1 \times 10^6$ cells) with 2 ml of DMEM/HIGH GLUCOSE medium supplemented with 10%

FBS. The culture medium was replaced with 2 ml of $Zn_{0.5}Fe_{2.5}O_4/SiO_2$ dispersed in DMEM (supplemented with 10% of FBS), at a concentration of 400 µg ml$^{-1}$. With different time gradients (0.5, 2, 6, 10 h), the culture dishes were exposed to the AMF ($H = 10$–$14\,kAm^{-1}$, $f = 430\,kHz$) to maintain the temperature between 43 and 45°C. The hyperthermia measurements in this study were done by using a HYPER5 machine fabricated by the MSI Company. A fluro-optic thermometer fibre probe (Neoptix Corp., CA) was used to probe the temperature every 1 s after switching on the magnetic applied field. The doping medium was removed after hyperthermia and the cell layer was washed with PBS, detached with 2.5% trypsin-EDTA solution and collected by centrifugation. The cell mass was resuspended with DMEM/HIGH GLUCOSE medium containing 10% CCK-8 solution at a density of $1 \times 10^5$ and transferred to 96-well plates ($n = 5$). After 1 h incubation, the efficiency of intracellular hyperthermia was assessed by the Cell Counting Kit-8 (CCK-8, Dojindo, Japan) assay. Human osteosarcoma MG-63 cells without NPs or AMF were used as the blank control group, human osteosarcoma MG-63 cells with NPs but without AMF were used as the other control group. The relative cell viability was calculated as the percentage of the blank control group.

### 2.5.2. Cell apoptosis analysis by flow cytometry

The pathways of cell death were investigated by staining with an Annexin-V-FITC/PI Apoptosis Detection Kit (Dojindo, Japan) and estimated by an FACS Calibur flow cytometer (Becton Dickinson, USA). In this experiment, human osteosarcoma MG-63 cells without NPs or AMF were used as a blank control group, human osteosarcoma MG-63 cells with NPs but without AMF were used as a negative control group, and human osteosarcoma MG-63 cells with NPs and exposed to AMF were used as a treatment group. Human osteosarcoma MG-63 cells were plated in a 35 mm culture dish and treated with AMF ($H = 10$–$14\,kAm^{-1}$, $f = 430\,kHz$) 43 and 45°C for 10 min. After treatment, all cells of each group were detached by the 2.5% trypsin solution and collected by centrifugation at 1000 rpm for 5 min, then washed twice with cold PBS (PH = 7.4). The cells were resuspended in 500 µl $1 \times$ Annexin-V Binding Buffer, then 100 µl of the cell solution was stained by both Annexin-V-FITC (5 µl) and PI (5 µl) in dark condition. After a dyeing period of 15 min, 400 µl $1\times$ Annexin-V Binding Buffer was added into the tube and the suspended cells were directly measured by a flow cytometer. For each group, $5 \times 10^5$ cells were counted and distinguished as living cells (Annexin-V-FITC-/PI−, lower left quadrant); early-stage apoptosis cells (Annexin-V-FITC+/PI−, lower right quadrant); late-stage apoptosis cells (Annexin-V-FITC+/PI+, upper right quadrant) and necrosis cells (Annexin-V-FITC-/PI+, upper left quadrant).

### 2.5.3. Immunofluorescence

The fluorescence image of human osteosarcoma MG-63 cells was observed by a confocal laser scanning microscopy (CLSM; TCS SP, Leica, Germany). The culture medium was discarded after magnetic hyperthermia, human osteosarcoma MG-63 cell layers were rinsed with cold PBS for three times. Then the cells were fixed with paraformaldehyde (4%) for 20 min in the incubator. Paraformaldehyde was removed and the cells were treated with Triton X-100 (Sigma-Aldrich, USA) for 30 min. After that, the human osteosarcoma MG-63 cells were stained by 4′,6-diamidino-2-phenylindole (Beyotime, Beijing, China. Excitation/emission: 364/454 nm) solution (2 µg ml$^{-1}$) for 5 min. The cells were imaged by CLSM for each group with the same settings.

# 3. Results and discussion

## 3.1. Characterization of $Zn_{0.5}Fe_{2.5}O_4/SiO_2$

In this study, $Zn_{0.5}Fe_{2.5}O_4$ NPs were synthesized by a one-pot approach. These zinc doping IONP were selected as a magnetic core for their outstanding balance between magnetic properties, surface-to-volume ratio for functionalization and proven biocompatibility [29]. The effect of zinc content on heating efficiency was investigated in our previous study [28]. Meanwhile, the size and shape are also important factors that may affect the heating efficiency. Seung-hyun Noh's group [30] revealed that core–shell cube (CS-cube) NPs lead to a large heat emission capability, thanks to their minimized surface anisotropy, reduced spin disordering and addition of exchange anisotropy. Guardia et al. [31] studied superparamagnetic iron oxide nanoparticles (SPIONs) with different sizes and found that SPIONs with an average diameter of $19 \pm 3$ nm had significant SAR values in the clinical condition. However, the naked NPs can easily aggregate when they are directly exposed to

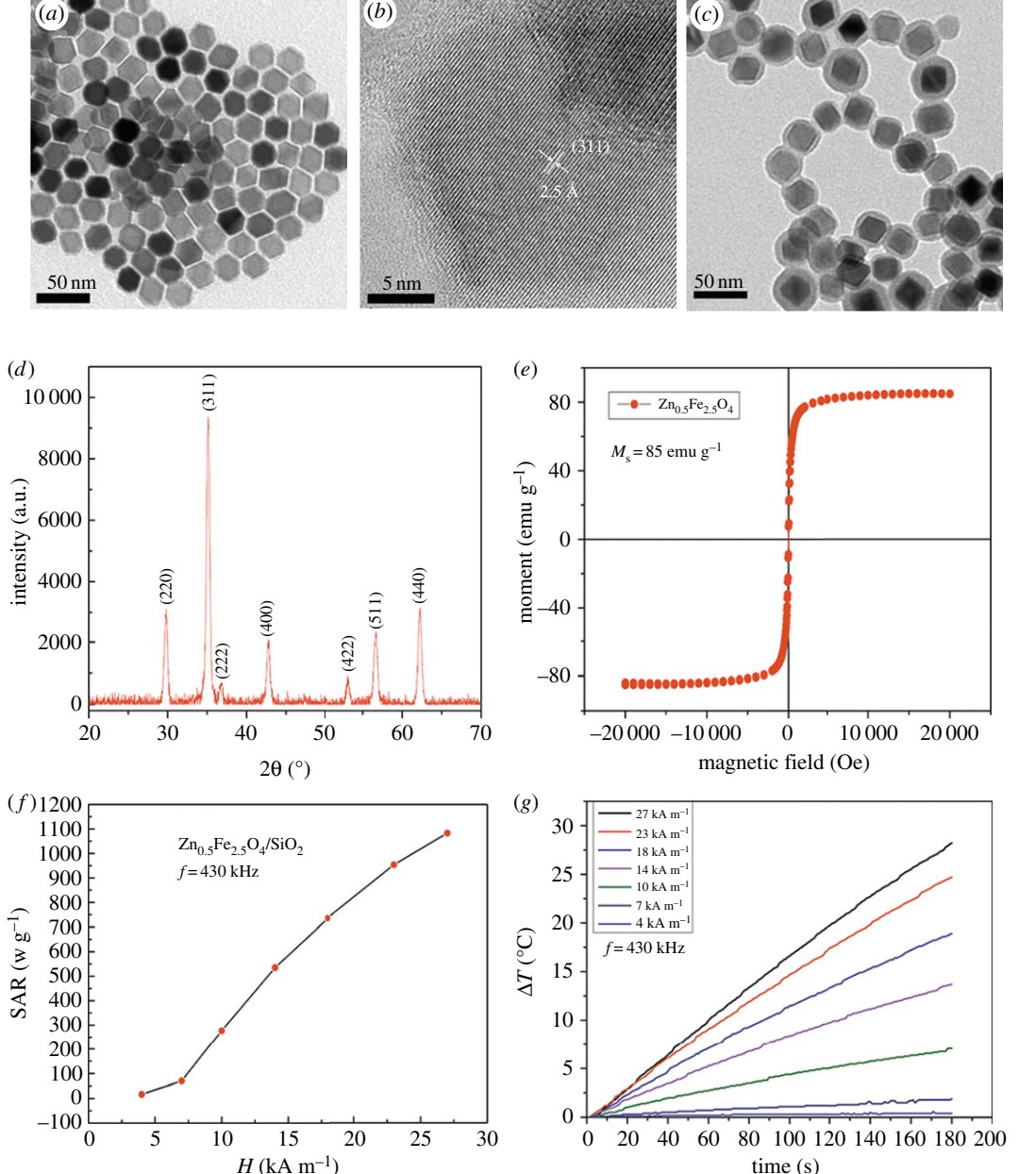

**Figure 1.** (*a*) TEM micrographs of $Zn_{0.5}Fe_{2.5}O_4$ NPs. (*b*) High-resolution TEM image of $Zn_{0.5}Fe_{2.5}O_4$ NPs. (*c*) TEM micrographs of silica-coated $Zn_{0.5}Fe_{2.5}O_4$ NPs. (*d*) XRD pattern of $Zn_{0.5}Fe_{2.5}O_4/SiO_2$ NPs, the main reflections are (220), (311), (222), (400), (511), (440) according to PDF#86-0509-jade. (*e*) Hysteresis loop for 22 nm $Zn_{0.5}Fe_{2.5}O_4/SiO_2$ NPs recorded at room temperature pointing to the superparamagnetic characteristics of NPs. (*f*) Field dependence of SAR for $Zn_{0.5}Fe_{2.5}O_4/SiO_2$ NPs. (*g*) Heating curves of aqueous solutions of 22 nm $Zn_{0.5}Fe_{2.5}O_4/SiO_2$ NPs (1 mg NPs ml$^{-1}$).

biological systems and react with oxygen in the air. Subsequently, silica was added to the solution to provide a surface coating with good biocompatibility and cellular internalization properties. Amorphous silica is approved by the United States Food and Drug Administration (US FDA) as a food additive, whereas crystalline silica is a suspected human carcinogen and is involved in the pathogenesis of silicosis [24].

The core–shell $Zn_{0.5}Fe_{2.5}O_4/SiO_2$ NPs consist of a magnetic core of $21.7 \pm 1.9$ nm and a silica-coated layer of $5.97 \pm 1.8$ nm, as observed in figure 1*a,c*. The average diameter of the NPs and coating estimated through the statistical analysis of TEM images (figure 1*a,c*) resulted in an average value of $21.7 \pm 1.9$ nm in good agreement with XRD values (figure 1*d*). A thin and uniform layer of silica coating was visible in the correlative TEM image (figure 1*c*), presenting their homogeneity in size and successful modification with no agglomeration. From the XRD diffractogram to PDF#86-0509-jade, the formation of the characteristic

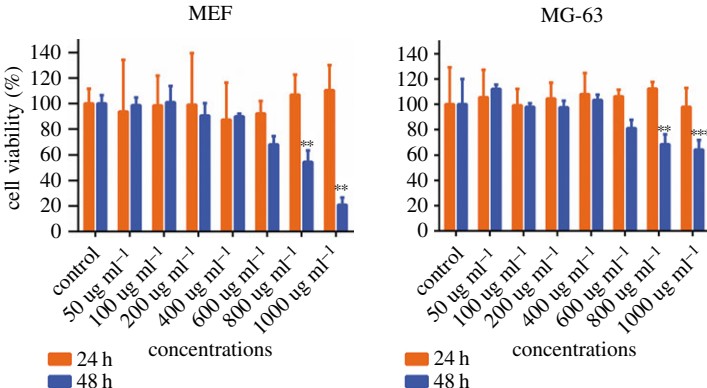

**Figure 2.** The cytotoxicity of $Zn_{0.5}Fe_{2.5}O_4/SiO_2$ NPs to MEF (*a*) and human osteosarcoma MG-63 (*b*) cells was assessed by the CCK-8 assay. Cells without NPs were used as control groups. Concentration-dependent cytotoxic effects of NPs were evaluated after 24 and 48 h incubation. Results are represented as mean ± s.e.m. Note: \*\*significant difference from control ($p < 0.01$); \*\*\*significant difference from control ($p < 0.005$).

spinel structure was confirmed (figure 1*d*). Electronic supplementary material, figure S1 shows that the core–shell $Zn_{0.5}Fe_{2.5}O_4/SiO_2$ NPs did not present significant changes in their hydrodynamic diameter even after keeping them in the aqueous solution for 72 h. Electronic supplementary material, figure S2 shows that the zeta potential (electronic supplementary material, figure S2a) of $Zn_{0.5}Fe_{2.5}O_4/SiO_2$ NPs is about −20 mV and the average size (electronic supplementary material, figure S2b) of NPs in aqueous dispersion is about 158.3 ± 41.2 nm, demonstrating their colloidal stability, which is vital when they are used as magnetic fluids in tumour hyperthermia.

The magnetic characteristic of $Zn_{0.5}Fe_{2.5}O_4/SiO_2$ NPs is critical for their biomedical application, especially in magnetic hyperthermia treatment. The saturation magnetization ($M_s$) of zinc ferrite is sensitive to Zn content, with $ZnFe_2O_4$ being antiferromagnetic $Zn^{2+}$ ions occupying the A-site of the spinel lattice only. The $M_s$ and anisotropy can be tuned by varying the Zn : Fe ratio [28], so the $M_s$ can be increased by doping zinc in ferrite properly. The $M_s$ of $Zn_{0.5}Fe_{2.5}O_4/SiO_2$ NPs was investigated by using VSM at room temperature (300 K), the magnetization value is 85 emu$^{-1}$ Fe. As shown in figure 1*e*, the hysteresis loop of NPs showed no hysteresis and obvious superparamagnetism. In addition, the $M_s$ of magnetite will be increased by substituting some $Fe^{2+}$ ions with $M^{2+}$ ($MFe_2O_4$, where M = Mn, Zn) in the magnetite structure. Owing to high biocompatibility and low toxicity, $Zn^{2+}$ can substituted in the place of $Fe^{2+}$ [32]. It is worthy to mention that the high $M_s$ value of the NPs tended to achieve distinct heating capability (figure 1*g*). The SAR can reach 1083 wg$^{-1}$ in AMF ($H = 27$ kAm$^{-1}$, $f = 430$ kHz) (figure 1*e*), which is sufficient for hyperthermia in clinical applications.

## 3.2. Biocompatibility of $Zn_{0.5}Fe_{2.5}O_4/SiO_2$

Cytotoxicity of $Zn_{0.5}Fe_{2.5}O_4/SiO_2$ NPs to cells was measured, the cell viability of MEF and human osteosarcoma MG-63 cells was assessed by the CCK-8 assay after incubation with various concentrations (50, 100, 200, 400, 600, 800, 1000 μg ml$^{-1}$) of NPs for 24 and 48 h. As shown in figure 2, the results demonstrated that MEF and human osteosarcoma MG-63 cells exposed to $Zn_{0.5}Fe_{2.5}O_4/SiO_2$ NPs resulted in time-dependent and concentration-dependent cytotoxicity. There was no statistical difference ($p > 0.05$) of cell viability between treatment and control groups after 24 h incubation, even the concentration was up to 1000 μg ml$^{-1}$. Although the viability of cells with $Zn_{0.5}Fe_{2.5}O_4/SiO_2$ is lower than that of control groups ($p < 0.01$) at 800 μg ml$^{-1}$ after 48 h incubation, it can still reach 54.6% for MEF and 68.7% for human osteosarcoma MG-63 cells.

Biocompatibility is essential for biomedical applications of MNPs [33]. The cytotoxicity of NPs is generally related to the structure and coating materials of NPs. IONPs are generally considered as safe, biocompatible and non-toxic materials [33]. However, biomedical applications of IONP doped with magnetically susceptible elements (e.g.$MnFe_2O_4$ and $CoFe_2O_4$) are much more restricted because of their potential toxicity and rapid oxidation, even though the magnetism of these nanoparticles is stronger than that of pure IONPs. Meanwhile, zinc is considered as an essential element for human beings, our previous study has also demonstrated that zinc doping IONPs are biocompatible and of low toxicity [28]. Furthermore, the biocompatibility depended on the coating materials [33]. Until now, only two families of

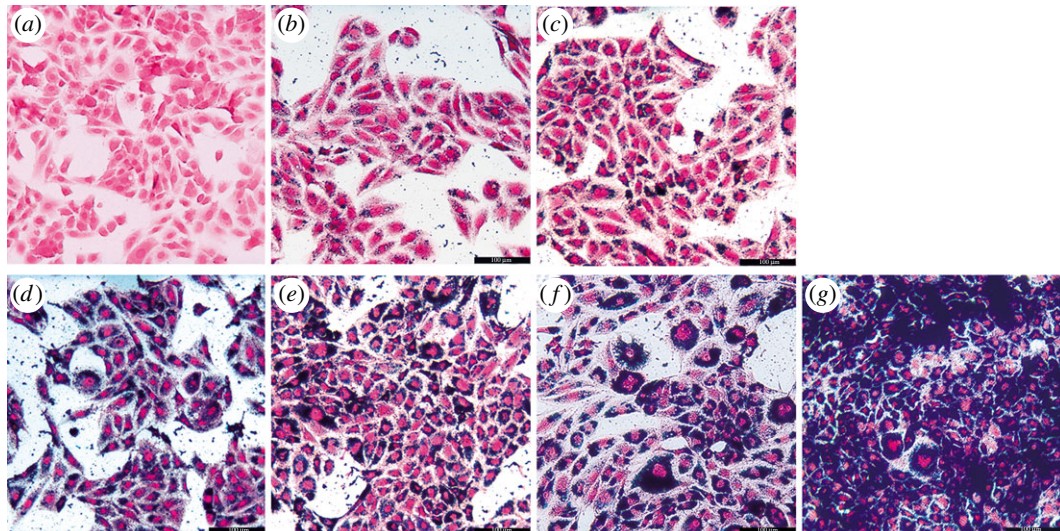

**Figure 3.** Human osteosarcoma MG-63 cells were incubated with various concentrations (*a*) control, (*b*) 6.25 µg ml$^{-1}$, (*c*) 12.5 µg ml$^{-1}$, (*d*) 25 µg ml$^{-1}$, (*e*) 50 µg ml$^{-1}$, (*f*) 100 µg ml$^{-1}$, (*g*) 200 µg ml$^{-1}$ of $Zn_{0.5}Fe_{2.5}O_4/SiO_2$ for 24 h. The human osteosarcoma MG-63 cells were stained blue at the concentration of 6.25 µg ml$^{-1}$ and the colour became thicker with the increase in NP concentrations.

IONPs have been applied in clinical trials, i.e. those coated with polysaccharides and those coated with silica [33]. Silica is 'generally recognized as safe' (GRAS) by the US FDA. Amorphous silica is used as an FDA-approved food additive [34]. The silica coating is a very common method for IONP coating because of its biocompatibility [35]. The silica coating can not only decrease the cytotoxicity for providing a stable protective layer against oxidation and reactive species [35] but also affect the heating efficiency. As shown in previous studies, the silica-coated NPs significantly decreased the SAR value by hindering the heat flow and the affection was related to the thickness of the silica layers. However, some studies demonstrated that the heating efficiency of the coated silica NPs was generally higher than that of bare magnetic core, Kaman *et al.* speculated that it was partially related to the different hydrodynamic behaviour, which represented that the silica-coated NPs possessed large hydrodynamic diameter and were strongly hydrated [36]. Although the optimal thickness of the coating is still an open question, Ansari [24] pointed out that the optimal amount of silica should be the minimum necessary to keep the core–shell NPs stable in water and also not to reduce the heat emission capability.

## 3.3. Cell uptake properties of $Zn_{0.5}Fe_{2.5}O_4/SiO_2$

### 3.3.1. Prussian blue staining

A detailed understanding of the cell uptake properties is critical when NPs are considered as magnetic hyperthermia agents. We have investigated two factors that may affect the cell uptake properties: the concentration of NPs and the incubation time. Some investigations demonstrated that temperature was also a factor that may affect the internalization. However, clinical application is the aim of magnetic hyperthermia.

Because human beings are homothermal (37°C), we think the temperature is not an essential factor. Firstly, human osteosarcoma MG-63 cells were incubated with various concentrations of $Zn_{0.5}Fe_{2.5}O_4/SiO_2$ NPs (6.25, 12.5, 25, 50, 100, 200 µg ml$^{-1}$) for 24 h (figure 3*a–f*), the cytoplasm was stained blue at the concentration of 6.25 µg ml$^{-1}$, with multiplication of concentrations, the blue colour became gradually thicker.

Figure 3 indicates that the internalization is related to the concentration of NPs. Secondly, human osteosarcoma MG-63 cells were incubated with 200 µg ml$^{-1}$ $Zn_{0.5}Fe_{2.5}O_4/SiO_2$ NPs for different times (0.5, 2, 4, 6, 8, 10, 12 h) (figure 4*b–h*). The cytoplasm was stained red in the control group, but the cytoplasm was stained blue after 0.5 h incubation. With the extension of the incubation time, the blue colour becomes gradually thicker. Figure 4 indicates that the internalization is related to the incubation time. In summary, the results demonstrated that the cell uptake properties of $Zn_{0.5}Fe_{2.5}O_4/SiO_2$ are concentration-dependent and time-dependent. The NPs could internalize into the tumour

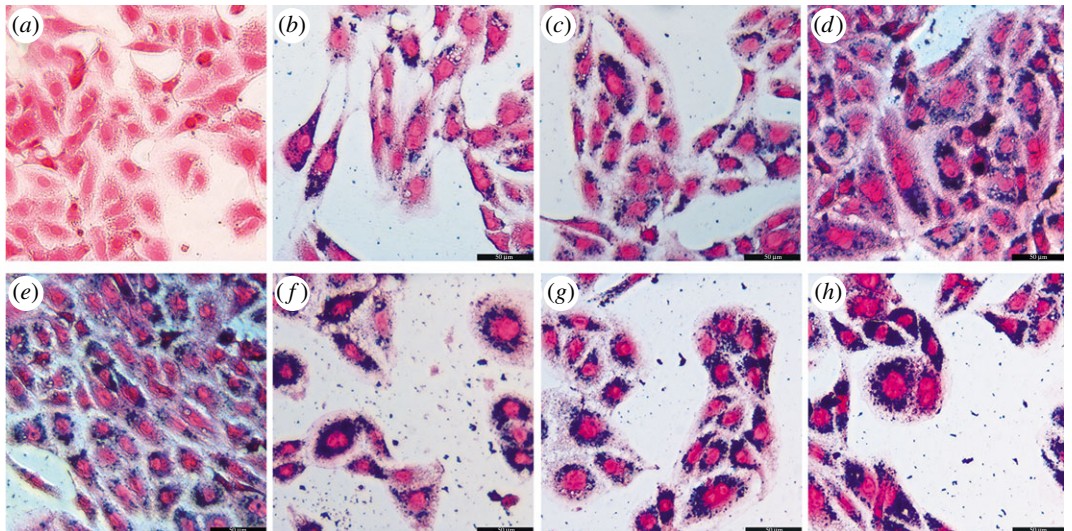

**Figure 4.** Human osteosarcoma MG-63 cells were incubated with 20 µg ml$^{-1}$ Zn$_{0.5}$Fe$_{2.5}$O$_4$/SiO$_2$ NPs for different times (($a$) control, ($b$) 0.5 h, ($c$) 2 h, ($d$) 4 h, ($e$) 6 h, ($f$) 8 h, ($g$) 10 h, ($h$) 12 h). The cytoplasm was stained blue at 0.5 h and became thicker with incubation time.

cells at an ultralow concentration (6.25 µg ml$^{-1}$), the internalization showed incubation at 0.5 h when the NP concentration was 200 µg ml$^{-1}$.

### 3.3.2. Transmission electron microscopy

TEM and Prussian blue staining are the essential methods to elucidate the location and fate of NPs in cells and tissues [25], so we study the cell uptake properties further by TEM. Figure 5$a$,$b$ show that the NPs just attached to the cellular membrane of the tumour cells after 0.5 h incubation, no internalization appeared at this time point. However, internalization was observed after 2 h incubation and the NPs were located in the lysosomes (figure 5$c$,$d$). With the extension of the incubation time, the amount of NPs was gradually increased. Plenty of lysosomes were loaded with NPs after 24 h incubation (figure 5$e$,$f$). Interestingly, the NPs showed good dispersion in the lysosomes at high-resolution TEM images (electronic supplementary material, figure S3), indicating that they were still able to heat up effectively and damage the tumour cells and thus they were still functional as hyperthermia agents [31]. Sadhukha [37] reported that well-dispersed NPs induced apoptosis, sub-micron size aggregation induced temperature-dependent autophagy by generating oxidative stress, micron size aggregation caused rapid membrane damage, resulting in acute cell kill. However, the acute cell killing and autophagy will induce acute inflammation, which is harmful to the periphric healthy tissues, so apoptosis is considered as the better tumour cell death pathway. In addition, Cabrera [27] revealed that aggregation affects NP magnetic characterization and heating efficiency. Therefore, NPs with good dispersion rather than aggregation are beneficial to keep heating efficiency inside cells. This section showed that Zn$_{0.5}$Fe$_{2.5}$O$_4$/SiO$_2$ can internalize into human osteosarcoma MG-63 cells, the internalization appeared about 0.5–2 h after incubation, TEM images showed that the NPs were all mostly located in lysosomes after endocytosis. It is worth noting that the NPs can still keep good dispersion in the lysosomes, which is beneficial to the heating emission efficiency.

### 3.3.3. Inductively coupled plasma–mass spectrometry

We have estimated the total amount of iron measured by ICP-MS. Human osteosarcoma MG-63 cells (1 × 10$^6$ ml$^{-1}$) were incubated with 200 µg ml$^{-1}$ Zn$_{0.5}$Fe$_{2.5}$O$_4$/SiO$_2$ NPs for different times (0.5, 2, 4, 6, 8, 10, 12 h). The results showed that the internalization increased with time. However, there was no statistical significant difference of the internalization between 2 h (14.23 Fe pg cell$^{-1}$) and 4 h (18.02 pg cell$^{-1}$) ($p >$ 0.05, $\alpha = 0.05$), but there was statistical significant difference among other time points ($p < 0.05$, $\alpha = 0.05$). Interestingly, the results were consistent with Prussian blue staining, which indicated time-dependent internalization. However, the TEM showed that NPs were just attached to the cytomembrane and did not

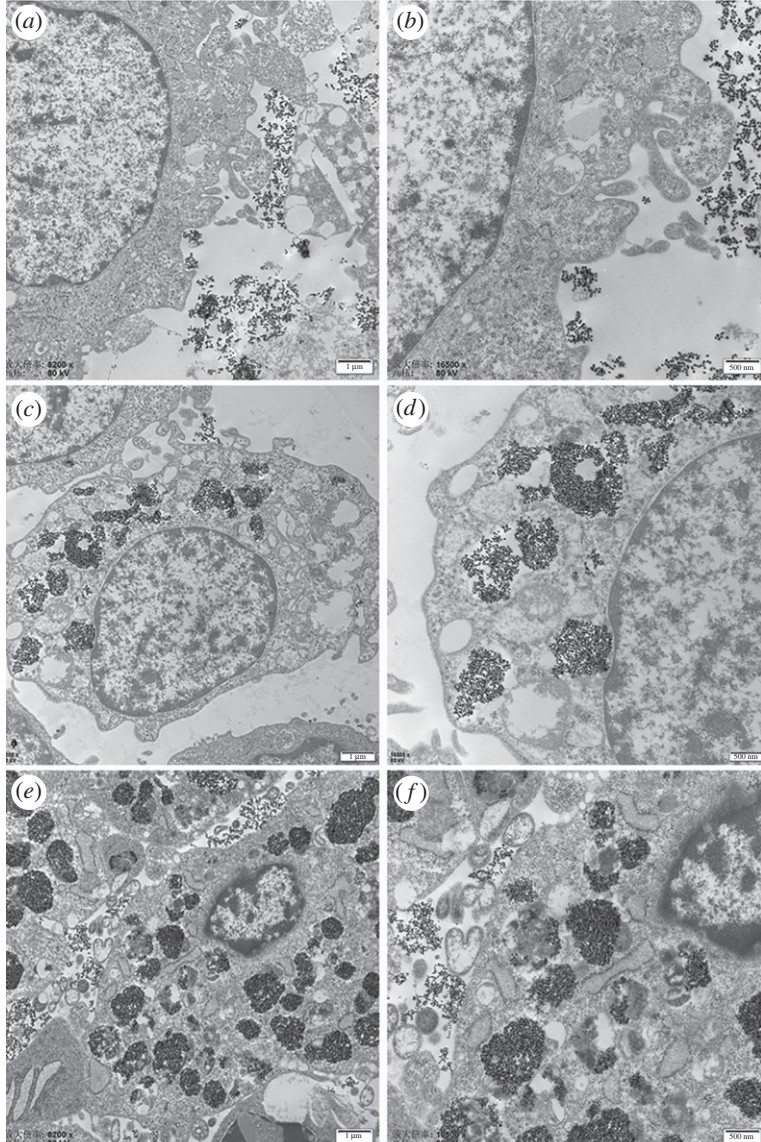

**Figure 5.** TEM characterization of human osteosarcoma MG-63 cell incubation with 200 µg ml$^{-1}$ Zn$_{0.5}$Fe$_{2.5}$O$_4$/SiO$_2$ NPs for 0.5 h (a,b), 2 h (c, d), 24 h (e,f). The NPs just attached to the cytomembrane did not internalize into the cytoplasm at 0.5 h incubation. However, the NPs internalized into human osteosarcoma MG-63 cells and were located in the lysosomes after 2 h incubation. With the extension of the incubation time, the amount of NPs in lysosomes gradually increased.

internalize into human osteosarcoma MG-63 cells at 0.5 h. We speculate the reason is that the extracellular NPs are difficult to rinse completely or some NPs are also on the cellular membrane.

## 3.3. Relationship of cell uptake and magnetic hyperthermia efficiency

Magnetic hyperthermia represents a promising therapeutic method to cancer treatment and is based on the mechanism that cancer cells are more sensitive than healthy cells to temperatures higher than 43°C [38]. Compared to various approaches proposed to raise the temperature [39], magnetic hyperthermia can offer several advantages [31]. One of the advantages is that magnetic hyperthermia can induce tumour cells to die without damaging the healthy tissues. The reason is that NPs can internalize into tumour cells and achieve 'inside-out' hyperthermia (intracellular hyperthermia). However, whether the cell killing efficiency will be enhanced with the increase of NPs internalized into the tumour cells, the best policy of internalization and the key mechanisms of NPs-meditated tumour cell death are still open questions. Recent studies showed that the mechanical rotation and/or the local temperature rise were the main mechanisms of internalized NPs-meditated tumour cell death [40], but which

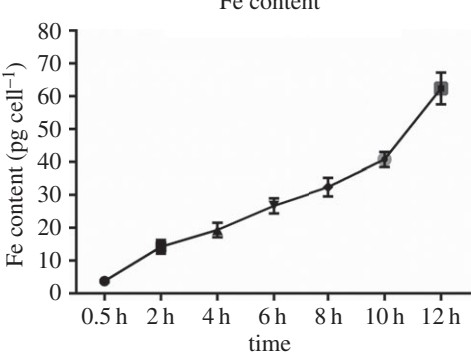

**Figure 6.** The amounts of iron per cell (in pg) determined by the elemental analysis are reported for human osteosarcoma MG-63 cell incubation with 200 µg ml$^{-1}$ Zn$_{0.5}$Fe$_{2.5}$O$_4$/SiO$_2$ for different times (0.5, 2, 4, 6, 8, 10, 12 h). These values indicate the amounts of iron internalized into the tumour cells. The iron contents were estimated by ICP-MS measurements of the treated cells. All points have been acquired in triplicate.

**Table 1.** Apoptotic assay of human osteosarcoma MG-63, induced by magnetic hyperthermia treatment of Zn$_{0.5}$Fe$_{2.5}$O$_4$/SiO$_2$ (no rinse group).

|  | control (%) | NPs (%) | 0.5 h (%) | 2 h (%) | 6 h (%) | 10 h (%) |
|---|---|---|---|---|---|---|
| Q1 | 0.058 | 0.015 | 1.89 | 11.3 | 5.26 | 6.57 |
| Q2 | 0.097 | 0.255 | 43.3 | 57.9 | 43.7 | 34.5 |
| Q3 | 0.335 | 3.5 | 17.7 | 8.13 | 9.54 | 6.83 |
| Q4 | 99.5 | 97.7 | 37.1 | 22.6 | 41.5 | 50.5 |

**Table 2.** Apoptotic assay of human osteosarcoma MG-63 induced by magnetic hyperthermia treatment of Zn$_{0.5}$Fe$_{2.5}$O$_4$/SiO$_2$ (rinse group).

|  | control (%) | NPs (%) | 0.5 h (%) | 2 h (%) | 6 h (%) | 10 h (%) |
|---|---|---|---|---|---|---|
| Q1 | 0.058 | 0.015 | 0.65 | 0.339 | 0.259 | 0.299 |
| Q2 | 0.097 | 0.255 | 5.25 | 5.67 | 5.49 | 7.56 |
| Q3 | 0.335 | 3.5 | 2.79 | 1.61 | 2.18 | 3.44 |
| Q4 | 99.5 | 97.7 | 91.3 | 92.4 | 92.1 | 88.7 |

mechanism plays a more important role? The literature does not show the answer, we think that it may depend on the structure of the NPs. Creixell *et al.* [17] synthesized epidermal growth factor receptor (EGFR)-targeted MNPs in AMF were shown to induce cancer cells significantly to die without a perceptible macroscopic temperature rise. Zhang *et al.* [16] covalently conjugated SPIONs with antibodies targeting the lysosomal protein marker LAMP1 (LAMP-SPIONs) and induced tumour cells to apoptosis through a tear of the lysosome membrane in a unique DMF. Maribella *et al.* [40] synthesized iron oxide MNPs targeted to the EGFR exposed to AMF that can selectively induce lysosomal membrane permeabilization in cancer cells over-expressing the EGFR, despite the SAR of these NPs being low (175 ± 18 W g$^{-1}$, $f$ = 233 kHz, $H$ = 40 kAm$^{-1}$). As mentioned above, these NPs were designed to induce tumour cells to die by the mechanical rotation. Meanwhile, there were also plenty of NPs designed to kill tumour cells by the local temperature rise [10,11,28]. Thus, we further investigate the relationship between the internalization and hyperthermia performance, then try to figure out which mechanism acts as the main role of Zn$_{0.5}$Fe$_{2.5}$O$_4$/SiO$_2$ NPs in inducing tumour cells to die.

Such Zn$_{0.5}$Fe$_{2.5}$O$_4$/SiO$_2$ NPs can be useful for magnetic hyperthermia because of their good biocompatibility and high SAR. We further investigate the hyperthermia performance of Zn$_{0.5}$Fe$_{2.5}$O$_4$/SiO$_2$ NPs *in vitro*. Cell apoptosis was examined by the flow cytometry based on the Annexin-V-FITC/PI Apoptotic Assay, as shown in tables 1 and 2 and figure 7a. Human osteosarcoma MG-63 cells

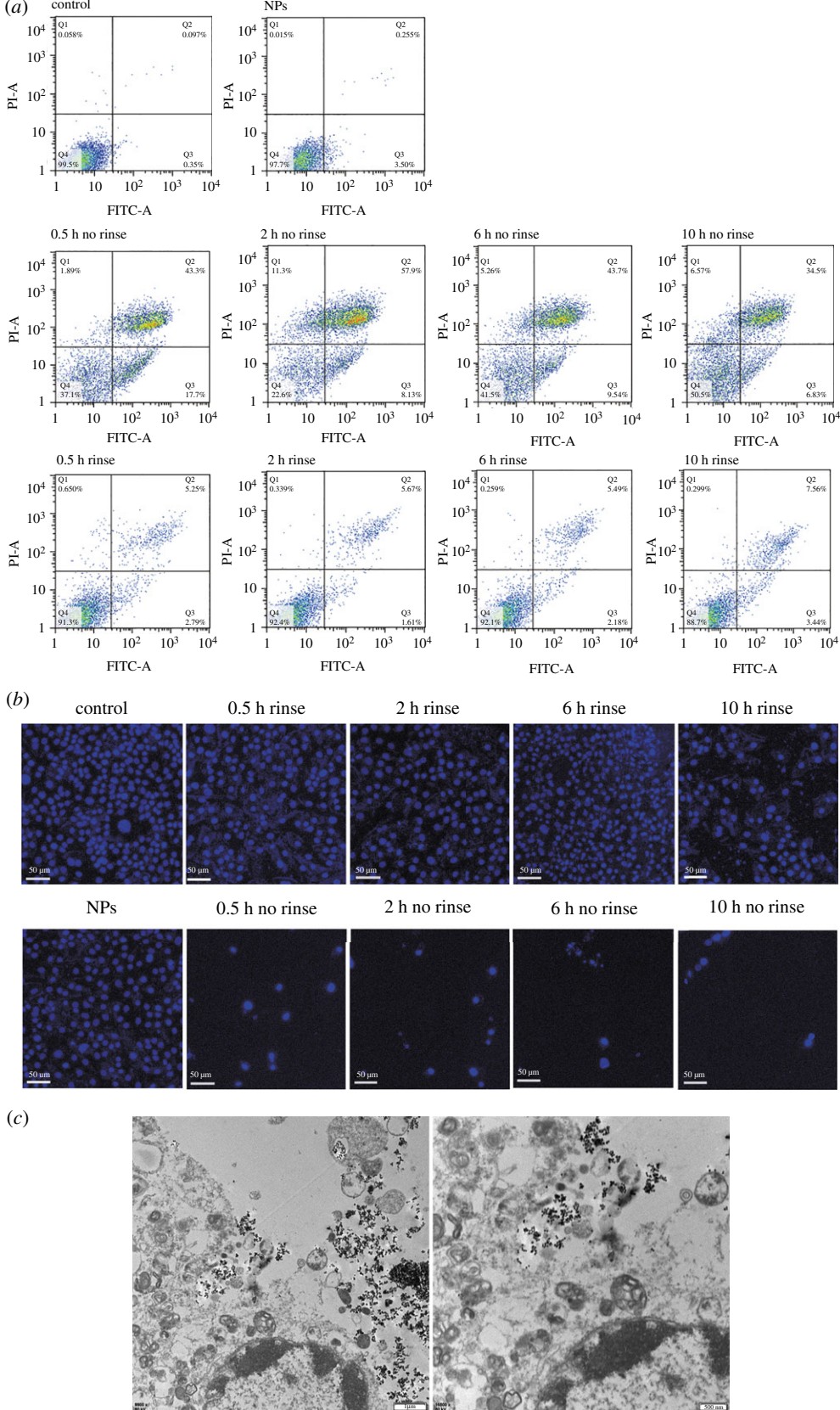

**Figure 7.** (*a*) Apoptotic assay of human osteosarcoma MG-63 cells incubated with $Zn_{0.5}Fe_{2.5}O_4/SiO_2$ for different times (0.5, 2, 6, 10 h) after being exposed to AMF ($f = 430$ kHz, $H = 11$ kAm$^{-1}$). Human osteosarcoma MG-63 cells without NPs or AMF were used as a control group. Human osteosarcoma MG-63 cells with NPs but without AMF were used as an another control group (NPs group). (*b*) The fluorescence imaging of human osteosarcoma MG-63 cells after magnetic hyperthermia treatment. (*c*) TEM imaging of human osteosarcoma MG-63 cells incubated with $Zn_{0.5}Fe_{2.5}O_4/SiO_2$ for 2 h after magnetic hyperthermia.

without NPs and human osteosarcoma MG-63 cells with NPs but without AMF were used as negative and positive control groups, respectively. Human osteosarcoma MG-63 cells with NPs and exposed to AMF (430 kHz, 11 kA m$^{-1}$) were the treatment groups. Each treatment group contained 'no rinse' and 'rinse' group, the 'rinse' group represented the extracellular NPs that were rinsed before hyperthermia, while the 'no rinse' group represented the extracellular NPs that were not rinsed before hyperthermia. These two types of treatment groups were then determined by the incubation time (0.5, 2, 6, 10 h). Tables 1 and 2 show that the percentage of the apoptotic cells of the 'no rinse' groups are obviously higher than the 'rinse' group at each time point. We consider the reason is that the concentration of NPs in the 'no rinse' group is higher than that of the 'rinse' group because the extracellular NPs are rinsed. So the heating efficiency of no rinse samples is higher than that of rinse samples. However, in 'no rinse' groups, the cell killing efficiency does not increase with the incubation time, the 0.5 and 2 h groups are higher than 6 and 10 h groups. The 2 h 'no rinse' group shows the highest cell apoptosis rate (66%), which is 61% in the 0.5 h group, 52% in the 6 h group and 41% in the 10 h group. Soukup *et al.* [19] demonstrated that the internalized NPs in live cells showed only Néel relaxation. As shown in figure 6, the internalized NPs of 6 and 10 h are more than 0.5 and 2 h ($p < 0.05$). Therefore, there are more NPs hindered in 6 and 10 h groups than 0.5 and 2 h groups, therefore the heating efficiency of 6 and 10 h is weaker than that of 0.5 and 2 h. In addition, in 'rinse' groups, although the cell killing efficiency does not show any pattern with the incubation time, the 10 h group shows the highest cell killing efficiency. As shown in figure 6, although the Brownian relaxation is hindered in all 'rinse' groups, the NP concentration of 10 h rinse samples is higher than that of the other 'rinse' groups. The tumour cells can be damaged by internalized NPs exposed to AMF by Néel relaxation. However, the Brownian relaxation will restore when the NPs come out from the ruptured tumour cells [19]. Therefore, the 10 h group shows the highest cell killing efficiency. It is worth mentioning that the late apoptosis is the main pathway of magnetic hyperthermia treatment in all groups. The results of flow cytometry are consistent with fluorescence imaging (figure 7b). The microstructure of human osteosarcoma MG-63 cells after magnetic hyperthermia treatment (figure 7c) showed that the NPs came out from the ruptured lysosomes and the cytomembrane was also damaged, but the chromatins showed edge aggregation, while the nuclear membranes were still intact. The features of TEM images further indicated that apoptosis was the main pathway to tumour cells, which was consistent with flow cytometry.

Our study demonstrated that $Zn_{0.5}Fe_{2.5}O_4/SiO_2$ NPs can be the candidate of magnetic hyperthermia agents. The NPs-meditated hyperthermia induced tumour cells to die at low concentration. However, the hyperthermia efficiency did not increase with the amount of internalized NPs in tumour cells. It is worth mentioning that the cell apoptosis rate of 2 h 'no rinse' is also higher than the 0.5 h 'no rinse' group. According to the TEM imaging (figure 5), we have known that the NPs attached just to the cytomembrane and do not internalize into tumour cells at 0.5 h incubation. Therefore, the best policy to enhance hyperthermia efficiency of NPs is not to saturate, but an appropriate amount of NPs are internalized into cells. Interestingly, although the extracellular NPs were rinsed, tumour cells could still be killed by NPs in the lysosomes. However, the cell kill efficiency of the 'rinse' group is obviously lower than that of the 'no rinse' group, and we still can monitor the macroscopic temperature rise (below 40°C). Hence, we speculate that the local temperature rise is the main role that induces tumour cells to die.

# 4. Conclusion

We have successfully synthesized monodispersed $Zn_{0.5}Fe_{2.5}O_4$ NPs of 22 nm by a one-pot approach and coated with silica as magnetic hyperthermia agents. The $Zn_{0.5}Fe_{2.5}O_4/SiO_2$ NPs have been characterized as superparamagnetic materials with high SAR and $M_s$. The results have shown that their excellent colloidal stability and low cytotoxicity can be considered as a magnetic fluid hyperthermia candidate. The investigation of cell uptake properties has demonstrated that such NPs can be internalized into human osteosarcoma MG-63 cells, the internalization appeared between 0.5 and 2 h and the NPs mostly located in lysosomes after internalization. Furthermore, the hyperthermia performance is related to the amount of internalized NPs, but the best amount is appropriate rather than saturated, which is worth studying further. The NPs can still induce tumour cell to die when extracellular NPs were rinsed. However, the local temperature rise is considered as the main role that induces tumour cell to die. These results inspire us to connect these NPs with special organella-targeted agents that will enhance both the temperature and mechanical mechanism in the future.

Data accessibility. The electronic supplementary material and original experiment data of this study are available within the Dryad Digital Repository: https://dx.doi.org/10.5061/dryad.sr0g5s1 [41].

Authors' contributions. K.M. and P.T. designed the study. R.W. and Y.L. accomplished whole cell experiment. J.L., R.Z. and Q.L. collected and analysed the data. X.Y., L.Z. and C.L. were responsible for the synthesis and coating of magnetite nanoparticles. R.W. interpreted the results and wrote the manuscript. All authors gave final approval for publication.

Competing interests. The authors declare no competing interests.

Funding. This work was supported by National Natural Science Foundation of China (grant nos 51772328, 81702121) and Natural Science Foundation of Guangxi Zhuang Autonomous Region (grant no. 2018JJB140367)

Acknowledgements. We are very grateful to Prof. Shuli He and the staff of Department of Physics of Capital Normal University for their guidance in the synthesis of nanoparticles. We thank Lin Chen, Shuolong Yuan and Jin Li of Translational Medical Center of PLA General Hospital for their careful laboratory assistance.

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
