## [Reviewer comments · Royal Society Open Science]

Review History

RSOS-191139.R0 (Original submission)

Review form: Reviewer 1

Is the manuscript scientifically sound in its present form?

Yes

Are the interpretations and conclusions justified by the results?

Yes

Is the language acceptable?

No

Do you have any ethical concerns with this paper?

No

Have you any concerns about statistical analyses in this paper?

No

Recommendation?

Accept with minor revision (please list in comments)

Comments to the Author(s)

This is an interesting work with a lot of experimental validation. My only concerns are that some interpretations require more sound justifications. See attached file (manuscript pdf file with comments (Appendix A)) for more details.

For example, what is the beneficiary role of Zn substitute to this specific MNPs

Why 2h incubation is the optimum time for internalization?

Hyperthermia interpretation requires more careful comments, than the general one that Brownian relaxation is hindered.

Why rinse and no rinse samples exhibit distinctly different features?

Finally, I have located a lot of grammar mistakes. The manuscript should undergo a careful English grammar and syntax checking by a language expert.

Review form: Reviewer 2**Is the manuscript scientifically sound in its present form?**

Yes

Are the interpretations and conclusions justified by the results?

Yes

Is the language acceptable?

No

Do you have any ethical concerns with this paper?

No

Have you any concerns about statistical analyses in this paper?

No

Recommendation?

Major revision is needed (please make suggestions in comments)

Comments to the Author(s)

I have no doubt that the paper is interesting. But I am also convinced that it needs a lot of editing, as the language is unclear in many places. A thorough revision of English is first of all recommended. Other observations follow:

1. Page 3, line 18: What is the meaning of DMF abbreviation?
2. Page 3, line 55: although this is provided later in the paper, some information regarding toxicity of Zn would be of interest. Why silica?
3. P6 L4: country of MSI company?
4. P 6, L 24: is this field-frequency combination within admitted limits?
5. P8 L3: 6.0+-1.8
6. L8 P28: the value of Ms is not large enough for justifying the huge SAR
7. The results in Fig 2 seem to indicate cytotoxicity

8. Fig 4: are the particle INSIDE or AROUND the cells?
9. P12 L56: not so novel technique. Also: this a very interesting part of the paper, but very unclear because of the language. Special care is suggested here

Decision letter (RSOS-191139.R0)

28-Sep-2019

Dear Professor Mao,

The editors assigned to your paper ("The cell uptake properties and hyperthermia performance of Zn_{0.5} Fe_{2.5}O₄ /SiO₂ nanoparticles as magnetic hyperthermia agents") have now received comments from reviewers. We would like you to revise your paper in accordance with the referee and Associate Editor suggestions which can be found below (not including confidential reports to the Editor). Please note this decision does not guarantee eventual acceptance.

Please submit a copy of your revised paper before 21-Oct-2019. Please note that the revision deadline will expire at 00.00am on this date. If we do not hear from you within this time then it will be assumed that the paper has been withdrawn. In exceptional circumstances, extensions may be possible if agreed with the Editorial Office in advance. We do not allow multiple rounds of revision so we urge you to make every effort to fully address all of the comments at this stage. If deemed necessary by the Editors, your manuscript will be sent back to one or more of the original reviewers for assessment. If the original reviewers are not available, we may invite new reviewers.

- Data accessibility

It is a condition of publication that all supporting data are made available either as supplementary information or preferably in a suitable permanent repository. The data accessibility section should state where the article's supporting data can be accessed. This section

should also include details, where possible of where to access other relevant research materials such as statistical tools, protocols, software etc can be accessed. If the data have been deposited in an external repository this section should list the database, accession number and link to the DOI for all data from the article that have been made publicly available. Data sets that have been deposited in an external repository and have a DOI should also be appropriately cited in the manuscript and included in the reference list.

If you wish to submit your supporting data or code to Dryad (<http://datadryad.org/>), or modify your current submission to dryad, please use the following link:
<http://datadryad.org/submit?journalID=RSOS&manu=RSOS-191139>

- **Competing interests**

- **Authors' contributions**

- **Acknowledgements**

- **Funding statement**

on behalf of Prof Pietro Cicuta (Subject Editor)
openscience@royalsociety.org

Associate Editor's comments:

Thank you for the submission. The reviewers identify a number of concerns with your paper that need to be addressed before publication may be considered. A key theme across both reports is that the quality of the writing is not of an appropriate standard - while perfect English isn't necessary, it seems the scientific story you are reporting is being obscured by the quality of the writing. With this in mind, you must seek advice from a service such as <https://royalsociety.org/journals/authors/language-polishing/> to help improve the English. Publication will be contingent on your satisfying the reviewers that the paper is of a suitable standard.

Comments to Author:

Reviewers' Comments to Author:

Reviewer: 1

Comments to the Author(s)

This is an interesting work with a lot of experimental validation. My only concerns are that some interpretations require more sound justifications. See attached file (manuscript pdf file with comments) for more details.

For example, what is the beneficiary role of Zn substitute to this specific MNPs

Why 2h incubation is the optimum time for internalization?

Hyperthermia interpretation requires more careful comments, than the general one that Brownian relaxation is hindered.

Why rinse and no rinse samples exhibit distinctly different features?

Finally, I have located a lot of grammar mistakes. The manuscript should undergo a careful English grammar and syntax checking by a language expert.

Reviewer: 2

Comments to the Author(s)

I have no doubt that the paper is interesting. But I am also convinced that it needs a lot of editing, as the language is unclear in many places. A thorough revision of English is first of all recommended. Other observations follow:

1. Page 3, line 18: What is the meaning of DMF abbreviation?
2. Page 3, line 55: although this is provided later in the paper, some information regarding toxicity of Zn would be of interest Why silica?
3. P6 L4: country of MSI company?
4. P 6, L 24: is this field-frequency combination within admitted limits?
5. P8 L3: 6.0+-1.8
6. L8 P28: the value of Ms is not large enough for justifying the huge SAR
7. The results in Fig 2 seem to indicate cytotoxicity
8. Fig 4: are the particles INSIDE or AROUND the cells?
9. P12 L56: not so novel technique. Also: this is a very interesting part of the paper, but very unclear because of the language. Special care is suggested here

Author's Response to Decision Letter for (RSOS-191139.R0)

See Appendix B.

RSOS-191139.R1 (Revision)

Review form: Reviewer 2

Is the manuscript scientifically sound in its present form?

Yes

Are the interpretations and conclusions justified by the results?

Yes

Is the language acceptable?

Yes

Do you have any ethical concerns with this paper?

No

Have you any concerns about statistical analyses in this paper?

No

Recommendation?

Accept as is

Comments to the Author(s)

The authors have made a significant effort in improving the MS in view of the reviewers' comments. The paper is now publishable

Decision letter (RSOS-191139.R1)

14-Nov-2019

Dear Professor Mao,

It is a pleasure to accept your manuscript entitled "The cell uptake properties and hyperthermia performance of Zn_{0.5}Fe_{2.5}O₄/SiO₂ nanoparticles as magnetic hyperthermia agents" in its current form for publication in Royal Society Open Science. The comments of the reviewer(s) who reviewed your manuscript are included at the foot of this letter.

Kind regards,

on behalf of the Associate Editor, and Professor Pietro Cicuta (Subject Editor)
openscience@royalsociety.org

Associate Editor Comments to Author:

Congratulations - the reviewer who expressed the greater concerns during the review of your initial iteration is now satisfied by the changes made and the paper may be accepted for publication.

Reviewer comments to Author:

Reviewer: 2
Comments to the Author(s)

The authors have made a significant effort in improving the MS in view of the reviewers' comments. The paper is now publishable

Appendix A**ROYAL SOCIETY
OPEN SCIENCE****The cell uptake properties and hyperthermia performance of
Zn_{0.5} Fe_{2.5}O₄ /SiO₂ nanoparticles as magnetic
hyperthermia agents**

Journal:	Royal Society Open Science
Manuscript ID	RSOS-191139
Article Type:	Research
Date Submitted by the Author:	03-Aug-2019
Complete List of Authors:	Wang, Runsheng; Medical School of Chinese PLA; The Third Affiliated Hospital of Guangxi Traditional Chinese Medicine University Liu, Jianheng; Medical School of Chinese PLA Liu, Yihao; Medical School of Chinese PLA Zhong, Rui; Medical School of Chinese PLA Yu, Xiang; Capital Normal University Liu, Qingzu; Medical School of Chinese PLA Zhang, Li; Capital Normal University Lv, Chenhui; Capital Normal University Mao, Keya; Medical School of Chinese PLA, Tang, Peifu; Medical School of Chinese PLA
Subject:	cellular biology < BIOLOGY, Nanotechnology < PHYSICS
Keywords:	Magnetite nanoparticles, Magnetic hyperthermia, Endocytosis, Cell uptake
Subject Category:	Biochemistry & Biophysics

Author-supplied statements

Relevant information will appear here if provided.

Ethics

Does your article include research that required ethical approval or permits?:

This article does not present research with ethical considerations

Statement (if applicable):

CUST_IF_YES_ETHICS :No data available.

Data

It is a condition of publication that data, code and materials supporting your paper are made publicly available. Does your paper present new data?:

Yes

Statement (if applicable):

The supplementary information and original experiment data of this study are available within the Dryad Digital Repository: <https://doi.org/10.5061/dryad.sr0g5s1>.

Reviewer URL: <https://datadryad.org/review?doi=doi:10.5061/dryad.sr0g5s1>

Conflict of interest

I/We declare we have no competing interests

Statement (if applicable):

CUST_STATE_CONFLICT :No data available.

Authors' contributions

This paper has multiple authors and our individual contributions were as below

Statement (if applicable):

Keya Mao and Peifu Tang designed the study. Runsheng Wang and Yihao Liu accomplished whole cell experiment. Jianheng Liu, Rui Zhong and Qingzu Liu collected and analysed the data. Xiang Yu, Li Zhang and Chenhui Lv were responsible for synthesis and coating of magnetite nanoparticles. Runsheng Wang interpreted the results and wrote the manuscript. All author gave final approval for publication.

The cell uptake properties and hyperthermia performance of $\text{Zn}_{0.5}\text{Fe}_{2.5}\text{O}_4/\text{SiO}_2$ nanoparticles as magnetic hyperthermia agents

Runsheng Wang^{1,2, †}, Jianheng Liu^{1, †}, Yihao Liu¹, Rui Zhong¹, Xiang Yu³, Qingzu Liu¹, Li Zhang³, Chenhui Lv³, Keya Mao^{1, #}, Peifu Tang^{1, #}

¹ Medical School of Chinese PLA, Beijing, 100853, China.

² Department of Orthopedics, The Third Affiliated Hospital of Guangxi Traditional Chinese Medicine University, Liuzhou, Guangxi Zhuang Autonomous Region, 545001 China.

³ Department of Physics, Capital Normal University, Beijing, 100048, China.

Key words: Magnetite nanoparticles, Magnetic hyperthermia, Endocytosis, Cell uptake.

Abstract

$\text{Zn}_{0.5}\text{Fe}_{2.5}\text{O}_4$ nanoparticles (NPs) of 22nm are synthesized by one-pot approach and coated with silica for magnetic hyperthermia agents. The nanoparticles exhibit superparamagnetic characteristics, high specific absorption rate (SAR) (1083 wg^{-1} , $f=430 \text{ kHz}$, $H=27 \text{ kAm}^{-1}$), large saturation magnetization ($M_s=85 \text{ emug}^{-1}$), excellent colloidal stability and low cytotoxicity. The cell uptake properties have been investigated by Prussian blue staining, transmission electron microscope (TEM) and inductively coupled plasma mass spectrometer (ICP-MS), which resulted in time dependent as well as concentration dependent internalization. The internalization appeared between 0.5h to 2h, the NPs were mainly located in the lysosomes and kept good dispersion after incubation with human osteosarcoma MG-63 cells. Then, the relationship between cell uptake and magnetic hyperthermia performance was studied. Our results showed that the hyperthermia efficiency rest with the amount of internalized NPs, which was depended on the concentration and incubation time. Interestingly, the NPs could still induce tumor cells to apoptosis/necrosis when extracellular NPs were rinsed, but the cell kill efficiency is lower than no rinse groups, which indicate that local temperature rise rather than mechanical mechanism is the main factor that induce tumor cells death. Our findings suggest that this high SAR and biocompatible silica coating $\text{Zn}_{0.5}\text{Fe}_{2.5}\text{O}_4$ nanoparticles could serve as a new agents for magnetic hyperthermia.

1. Introduction

Magnetic nanoparticles have been attracted tremendous attention for their diverse biomedical applications including drug delivery[1], imaging[2], tumor targeting with various chemotherapeutic drugs[3], cell separation[4], gene therapy[5] and magnetic hyperthermia[6]. Hyperthermia is used as an adjunctive treatment in tumor therapy, the mechanism of hyperthermia is that tumor cells are much more sensitive to heat shock than normal healthy cells when exposed to temperature above $43-45^\circ\text{C}$, their proliferation and metabolic activity is inhibited which can lead to either apoptosis or necrosis[7]. Intracellular magnetic hyperthermia was firstly proposed by Gordon in

Author of correspondence

Keya Mao

e-mail:maokeya@sina.com

Peifu Tang

e-mail:pftang301@126.com

†These two authors contributed equally to this work.

1979[8], which had more advantages than conventional hyperthermia (such as, microwaves, radiofrequency and
high-intensity focused ultrasound) for resulting in heating the tumor cells from "inside-out" that can prevent the
unexpected damage to surrounding normal tissue[9]. Current studies have also investigated the excellent
hyperthermia performance of magnetic nanoparticles when internalized into tumor cells [10]. Afterwards kinds of
core-shell iron oxide nanoparticles have been designed to enhance the target to organelle of tumor cells, which can
improve hyperthermia performance [11,12]. Bi-functional even multi-functional nanoparticles have been designed
to trigger tumor cells to apoptosis or necrosis by alternating magnetic fields or lasers [13,14]. Although pulsed
magnetic fields can improve the transport of iron oxide nanoparticles through cell membrane which means
increasing the amount of cell internalization[15], some kinds of NPs even can motivate apoptosis or necrosis of
cancer cells without the rise of temperature in the special magnetic fields (such as DMF)[16,17]. Although the
designs of magnetic nanoparticles are different, the mechanisms of heating emission are the same. Two main
magnetic relaxation mechanisms determine the thermal efficiency of NPs, i.e., the magnetic process (Néel
relaxation) by which the magnetic moments fluctuate without NPs rotation, and the mechanical process (Brownian
relaxation) by which the NPs rotate with their magnetic moment fixed at a given crystal axis[18].
However, the mobility of NPs which is considered relating to Brownian relaxation will be inhibited when NPs
internalize into tumor cells, then the heating efficiency decreases due to the NPs can only undergo Néel
relaxation[19]. This phenomenon has been proved not only in vitro [20] but also in vivo[21] studies. Meanwhile, for
the safety of patients, the product of magnitude and frequency of alternating magnetic fields must be less than
$5 \times 10^9 \text{ Am}^{-1}\text{s}^{-1}$ [22]. Consequently, increasing NPs concentration may be the way to improve the hyperthermia
performance, but the long-term toxicity of nanoparticles is still unclear, previous studies have demonstrated that
the breakdown and clearance of heavy metals containing in nanoparticles is quite slow[23]. In summary, NPs with
high specific absorption ratio (SAR) are needed to reduce amount of particles delivered to the body with regard to
the as yet incompletely known toxicity. The higher SAR can partly compensate for the lower concentration of
particles in the tumor[19], while coated with silica can reduce the cytotoxicity and prevent NPs from aggregation
and oxidation[24].
Cell uptake is also called internalization or endocytosis, which is a cellular process in which substances are
penetrated into the cell. The material to be internalized is surrounded by an area of plasma membrane, which then
buds off inside the cell to form a vesicle containing the ingested material. Endocytosis includes pinocytosis (cell
drinking) and phagocytosis (cell eating). The interactions between NPs and cells are critical when NPs are
considered to biomedical applications[25]. Cell uptake properties are the crucial factors that may affect the
magnetic hyperthermia efficiency. The size and surface modifications of NPs lead to different responses in terms
of cell interaction.[26] Riccardo[20] et al revealed that the fall in heating efficiency correlated with a complete
inhibition of Brownian relaxation in cellular conditions. Then Soukup[19] et al verified the block of Brownian
relaxation through the AC susceptibility signal from internalized nanoparticles in live cells, which was consistent
with measurements of immobilized nanoparticles suspensions. However, Cabrera[27] et al suggest that the
enhancement of intracellular iron oxide nanoparticles (IONP) clustering mainly lead to the decrease of heating
efficiency rather than intracellular IONP immobilization. Hence, a detailed understanding cell uptake properties is
crucial when NPs are considered as hyperthermia mediators.
Here we synthesize high heating efficiency and low cytotoxicity silica coated zinc doping iron oxide nanoparticles,
the cell uptake properties and intracellular hyperthermia performance were investigated. Our investigation
demonstrates that $\text{Zn}_{0.5}\text{Fe}_{2.5}\text{O}_4/\text{SiO}_2$ nanoparticles exposed to human osteosarcoma MG-63 cells resulted in
time-dependent as well as concentration-dependent internalization. The internalization appeared between 0.5 to 2

hours after incubation. The intracellular hyperthermia efficiency was regard to the amount of NPs in tumor cells. It is worth mention that the tumor cells could also be killed when the extracellular NPs were rinsed. Our finding suggest that this high SAR and biocompatible silica coating $Zn_{0.5}Fe_{2.5}O_4$ nanoparticles could serve as a new agents for magnetic hyperthermia.

2. Methods

2.1. Nanoparticles synthesis and characterization

The synthesis methods of $Zn_{0.5}Fe_{2.5}O_4$ nanoparticles and silica coating had been reported before[28].

Materials

Zinc (II) acetylacetonate, iron (III) acetylacetonate and sodium oleate were purchased from Alfa-Aesar. Oleate (90%), oleic acid (90%), benzyl ether (98%), Igepal CO-520, cyclohexane, tetraethylorthosilicate (TEOS) and ammonium hydroxide were purchased from Sigma-Aldrich. Ethanol and hexane were purchased from Beijing Chemical Co., Ltd. (China).

Synthesis of $Zn_{0.5}Fe_{2.5}O_4$ nanoparticles

Zinc (II) acetylacetonate (0.5 mmol), iron (III) acetylacetonate (2.5 mmol), oleic acid (4 mL) and sodium oleate (2 mmol) were mixed with benzyl ether (20mL) under an argon flow. The solution was stirred by a magnetic stirring apparatus under argon atmosphere and heated to 393 K for 1 h. The mixture was further heated to reflux temperature ($\approx 573K$) and kept at this value for 1 h. After cooling down to room temperature, the NPs were collected by centrifugation. The size of magnetic NPs rested with the heating rate from 393 to 573 K.

Silica Coating of NPs

The hydrophobic NPs were turned to hydrophilic solutions by coating silica shells via a reverse microemulsion method. Cyclohexane (10 mL) and Igepal CO-520 (0.575 mL) were mixed by ultrasonic agitator for 10 min, then magnetic NPs (10 mg) in cyclohexane (1ml) were added and stirred for 30 min. Ammonium hydroxide(0.075ml,28-30%) was then added and followed by 0.05ml TEOS. The solution was sealed and stirred at room temperature for 24h. Then ethanol and hexane were added and the $Zn_{0.5}Fe_{2.5}O_4/SiO_2$ core/shell NPs were collected by centrifugation. Finally, the core/shell NPs were washed twice in ethanol and deionized water and collected by centrifugation and redispersed in deionized water.

Characterization

The morphology and size of NPs was characterized by a High resolution Hitachi H7650 (120kV) transmission electron microscope (TEM). The structural formations of NPs were determined by X-ray diffraction (XRD) (D8 ADVANCE, Bruker Corp., DE). A vibrating sample magnetometer (VSM) (3473-70, GMW Corp., NZ) was used to measure the magnetic hysteresis loops. To investigate the hyperthermia performance of $Zn_{0.5}Fe_{2.5}O_4/SiO_2$, the SAR was characterized by the alternating magnetic field (AMF) (HYPER5, MSI Corp., USA) with frequency of 430 kHz. A fiber optic probe (Neoptix Corp., CA) was applied to probe the temperature of NPs solution every 1s after turning on the magnetic field.

2.2. Cell culture

Mouse embryonic fibroblast (MEF) cells and human osteosarcoma MG-63 cells were all purchased from the American Type Culture Collection (ATTC, USA). Cells were all cultured in DMEM/HIGH GLUCOSE (Hyclone, USA) medium supplemented with 10% fetal bovine serum (FBS) and antibiotics (100 $\mu\text{g ml}^{-1}$ penicillin and 100 $\mu\text{g ml}^{-1}$ streptomycin) in a humidified atmosphere of 5% CO_2 at 37°C.

2.3. Cytotoxicity

Cell Counting Kit-8(CCK-8, Dojindo, Japan) assay was used to evaluate the cytotoxicity of $Zn_{0.5}Fe_{2.5}O_4 /SiO_2$ to human osteosarcoma MG-63 and MEF cells. For CCK-8 assay, the MEF and human osteosarcoma MG-63 cells were incubated in 96-well plates at a density of 1×10^4 per well and grown overnight ($n=5$ per group),and then co-incubated with various concentrations (0,25,50,100,200,400,800,1000 $\mu\text{g ml}^{-1}$) of $Zn_{0.5}Fe_{2.5}O_4 /SiO_2$ at 37°C for

24 h and 48h. Following this incubation, each well was washed with phosphate buffered saline (PBS) for 3 times.
Then the cells were incubated in media with 10% CCK-8 for 1h. After this incubation, the solution transferred to a
blank 96-well plate and the NPs were fixed with a permanent magnet under the bottom of the plate in order to
avoid the affection of NPs in the process of optical density (OD) measurement. The absorbance was measured at
450 nm with multimode plate reader (TECAN InfiniteM200 PRO, Switzerland). Cell viability was expressed as
the percentage of viable cells compared with controls (cells treated with culture medium).

2.4. Cell uptake

Prussian blue staining

The cellular uptake of $Zn_{0.5}Fe_{2.5}O_4/SiO_2$ nanoparticles to human osteosarcoma MG-63 cells were measured by
Prussian blue staining (Solarbio, Beijing, China). Various concentrations (6.25, 12.5, 25, 50, 100, 200ug/ml) of
$Zn_{0.5}Fe_{2.5}O_4/SiO_2$ nanoparticles were incubated with human osteosarcoma MG-63 cells in 24-well plates at a
density of 10^5 cells per well (n=4 per group). After 24 hours incubation, the cell layers were washed with
phosphate buffered saline (PBS) for 3 times, fixed with 4% paraformaldehyde for 10min in 4 °C. To stain the
intracellular iron, the Prussian blue solution mixed with of 2% hydrochloric acid aqueous solution and 2%
potassium ferrocyanide (II) trihydrate was incubated with the fixed cells for 30 min in 37°C. Then, the fixed cells
were washed with ultrapure water for 3 times and counterstained with nuclear eosin for 20 s. After washing with
ultrapure water, the cells were placed on a microscope for cellular uptake observation. $Zn_{0.5}Fe_{2.5}O_4/SiO_2$
nanoparticles (200ug ml⁻¹) were incubated with human osteosarcoma MG-63 at different
time(0.5h,2h,4h,6h,8h,10h,12h) in 24-well plates at a density of 10^5 cells per well (n=4 per group). After
incubation, the intracellular iron were also measured by standard Prussian blue staining as described previously.

TEM

TEM images were recorded with a FEI Tecnai G2 spirit microscope (Thermo Fisher Scientific, USA) operating at
an accelerating voltage of 120kV. For the characterization of the doped cells, after the incubation with $Zn_{0.5}Fe_{2.5}O_4$
$/SiO_2$ nanoparticles, the human osteosarcoma MG-63 cells (1×10^6) were washed three times with phosphate
buffered saline (PBS) and fixed with 1.5% glutaraldehyde in phosphate buffered saline (PBS) at 4°C for 30 min.
The fixed cells were washed three times with phosphate buffered saline (PBS) and then 1% osmium tetroxide in
phosphate buffered saline (PBS) was added for 1 h at room temperature. After another three washing steps in
phosphate buffered saline (PBS), the cells were dehydrated with 30, 50, 75, 85, 95, and 100% (three times)
absolute ethanol. Thereafter, the cells were infiltrated with Epon resin (two steps: 50 and 66% for resin in absolute
ethanol, 30 min each) and embedded in 100% resin at 60°C for 2 days. Ultrathin sections (70 nm thick) were cut on
an Ultramicrotome (Leika), stained with lead citrate, and observed by TEM.

ICP-MS

$Zn_{0.5}Fe_{2.5}O_4/SiO_2$ nanoparticles at concentration of 200ug ml⁻¹ were incubate with human osteosarcoma MG-63
cells (1×10^6) for different time (0.5, 2, 4, 6, 8, 10, 12h). Then, the extracellular NPs were rinsed with PBS for 3
41 times, the cells were detached with 2.5% trypsin-EDTA solution, and collected by centrifugation (1000rpm×5min).
The samples were digested in concentrated 3:1 HCl/HNO₃ (v/v) solutions, the amount of internalization iron was
determined by elemental analysis using an ICP-MS spectrometer (Thermo Fisher Scientific, USA). All time points
have been acquired in triplicate.

2.5. Intracellular hyperthermia

Cytotoxicity of IH

Human osteosarcoma MG-63 cells were plated in 35-mm inner diameter culture dishes at 50% confluence (1×10^6
cells) with 2 mL of DMEM/HIGH GLUCOSE medium supplemented with 10% FBS. The culture medium was
replaced with 2 mL of $Zn_{0.5}Fe_{2.5}O_4/SiO_2$ dispersed in DMEM (supplemented with 10% of FBS), at a concentration
of 400ugmL⁻¹. With different time gradient(0.5, 2, 4 ,6, 8, 10, 12), the culture disdes were exposed to the AMF (H

10-14kAm⁻¹, f 430kHz)to maintain the temperature between 43 to 45°C. The hyperthermia measurements in this study were performed by using a HYPER5 machine fabricated by MSI Company. A fluoro-optic thermometer fiber probe (Neoptix Corp., CA) was used to probe the temperature every 1 s after switching on the magnetic applied field. The doping medium was removed after hyperthermia and the cell layer was washed with PBS, detached with 2.5% trypsin-EDTA solution, and collected by centrifugation. The cell mass was resuspended with DMEM/HIGH GLUCOSE medium contain 10% CCK-8 solution at density of 1×10⁵ and transferred to 96 well plates (n=5) .After 1 hour incubation, the efficiency of IH was assessed by the Cell Counting Kit-8 (CCK-8, Dojindo, Japan) assay. Human osteosarcoma MG-63 cells without NPs or AMF were used as blank control group, human osteosarcoma MG-63 cells with NPs but without AMF were the other control group. The relative cell viability was calculated as the percentage of blank control group.

Cell apoptosis analysis by flow cytometry

The pathways of cell death were investigated by staining with Annexin-V-FITC/PI Apoptosis Detection Kit (Dojindo, Japan), and estimated by an FACS Calibur flow cytometer (Becton Dickinson, USA). In this experiment, human osteosarcoma MG-63 cells without NPs or AMF were used as a blank control group, human osteosarcoma MG-63 cells with NPs but without AMF were used as negative control group, human osteosarcoma MG-63 cells with NPs and exposed to AMF were used as treatment group. Human osteosarcoma MG-63 cells were plated in 35mm culture dish and treated with AMF (H 10-14kAm⁻¹, f 430 kHz) with temperature between 43 to 45°C for 10 min. After treatment, all cells of each group were detached by 2.5% trypsin solution and collected by centrifugation at 1000rpm for 5min, then washed twice with cold PBS (PH=7.4). The cells were resuspended in 500ul 1× Annexin V Binding Buffer, then 100ul of the cell solution was stained by both Annexin-V-FITC (5ul) and PI(5ul) in dark condition. After a dyeing period of 15 min, 400ul 1× Annexin V Binding Buffer was added into the tube and the suspended cells were directly measured by flow cytometer. For each group, 5 × 10⁵ cells were counted and distinguished as living cells (Annexin-V-FITC-/PI-, low left quadrant); early stage apoptosis cells (Annexin-V-FITC+/PI-, low right quadrant); late-stage apoptosis cells (Annexin-V-FITC+/PI+, upper right quadrant), and necrosis cells (Annexin-V-FITC-/PI+, upper left quadrant).

Immunofluorescence

The fluorescence imaging of human osteosarcoma MG-63 cells were observed by a confocal laser scanning microscopy (CLSM, TCS SP, Leica, Germany). The culture medium were discarded after magnetic hyperthermia, human osteosarcoma MG-63 cell layers were rinsed with cold PBS for three times. Then the cells were fixed with paraformaldehyde (4%) for 20 min in incubator. Paraformaldehyde was removed and the cells were treated with Triton X-100 (Sigma-Aldrich, USA) for 30 min. After that, the human osteosarcoma MG-63 cells were stained by DAPI (Beyotime, Beijing, China. Excitation/emission: 364/454 nm) solution (2ug ml⁻¹) for 5 min. The cells were imaged by CLSM for each groups with the same settings.

3. Results and Discussion

3.1. Characterization of Zn_{0.5}Fe_{2.5}O₄/SiO₂

Fig.1: (a) TEM micrographs of $Zn_{0.5}Fe_{2.5}O_4$ nanoparticles. (b) High resolution TEM image of $Zn_{0.5}Fe_{2.5}O_4$ nanoparticles. (c) TEM micrographs of silica coated $Zn_{0.5}Fe_{2.5}O_4$ nanoparticles. (d) XRD pattern of $Zn_{0.5}Fe_{2.5}O_4/SiO_2$ nanoparticles, the main reflections are (220), (311), (222), (400), (511), (440) according to PDF#86-0509-jade. (e) Hysteresis loop for 22nm $Zn_{0.5}Fe_{2.5}O_4/SiO_2$ nanoparticles recorded at room temperature pointing the Superparamagnetic characteristics of NPs. (f) Field dependence of SAR for $Zn_{0.5}Fe_{2.5}O_4/SiO_2$ nanoparticles. (g) Heating curves of aqueous solutions of 22nm $Zn_{0.5}Fe_{2.5}O_4/SiO_2$ nanoparticles ($1 mg_{NPs} ml^{-1}$).

In this study, $Zn_{0.5}Fe_{2.5}O_4$ nanoparticles were synthesized by one-pot synthesis approach. These zinc doping iron oxide nanoparticles were selected as magnetic core for their outstanding balance between magnetic properties, surface-to-volume ratio for functionalization and proven biocompatibility[29]. The affecton of zinc content to heating efficiency has been investigated in our previous study[28]. Meanwhile, the size and shape are also the important factors that may affect the heating efficiency. Seung-hyun Noh's group[30] revealed that core-shell cube(CS-cube) nanoparticles lead to a large heat emission capability, thanks to their minimized surface anisotropy, reduced spin disordering and additional of exchange anisotropy. Guardia et al[31] studied superparamagnetic iron oxide nanoparticles (SPIONs) with different sizes, they founded that SPIONs with an average diameter of $19 \pm 3 nm$ had significant SAR values in clinical condition. However, the naked form nanoparticles can easily form aggregates when they are directly exposed to biological systems and react with oxygen in the air. Subsequently, silica was added to the solution to provide a surface coating with good biocompatibility and cellular internalization properties. Amorphous silica was approved by US FDA as food additive, whereas crystalline silica is a suspected human carcinogen and is involved in the pathogenesis of silicosis[24].

The core-shell $Zn_{0.5}Fe_{2.5}O_4/SiO_2$ nanoparticles consist of a magnetic core of $21.7 \pm 1.9 nm$ and a silica coating layer

of 5.97 ± 1.8 nm were observed in Fig 1a and Fig 1c. Estimation of the average diameter of the nanoparticles and coating was performed through statistical analysis of Transmission electron microscopy (TEM) image (Fig 1a, Fig 1c) resulted to an average value of 21.7 ± 1.9 nm in good agreement with XRD values (Fig 1d). A thin and uniform layer of silica coating was visible in correlative TEM image (Fig 1c), presenting their homogeneity in size and successful modification with no agglomeration. From XRD diffractogram as compared to PDF#86-0509-jade, the formation of the characteristic spinel structure was confirmed (Fig 1d). Figure S1 showed that the core-shell $Zn_{0.5}Fe_{2.5}O_4/SiO_2$ nanoparticles did not present significant changes in their hydrodynamic diameter even after keeping in aqueous solution for 72h. Figure S2 showed that the zeta potential (Fig. S2 a) of $Zn_{0.5}Fe_{2.5}O_4/SiO_2$ nanoparticles is about -20 mV and the average size (Fig. S2 b) of NPs in aqueous dispersion is about 158.3 ± 41.2 nm, demonstrating their colloidal stability, which is vital when they were used as magnetic fluids in tumor hyperthermia.

The magnetic characteristic of $Zn_{0.5}Fe_{2.5}O_4/SiO_2$ nanoparticles is critical for their biomedical application, especially in magnetic hyperthermia treatment. The saturation magnetization (M_s) of $Zn_{0.5}Fe_{2.5}O_4/SiO_2$ nanoparticles was investigated by using VSM at room temperature (300K), showing a high magnetization value ($85 \text{ emu}^{-1} \text{ Fe}$). As showed in Fig 1e, the hysteresis loop of NPs showed no hysteresis and obvious superparamagnetism. In addition, the M_s of magnetite will be increased by substituting some Fe^{2+} ions with M^{2+} (MFe_2O_4 , where $M = Co, Mn, Ni, Zn$) in the magnetite structure. Due to high bio-compatibility and low toxicity, Zn^{2+} can be a proper ion to be substitute instead of Fe^{2+} . It was worthy to mention that the high M_s value of the NPs tended to achieve distinct heating capability (Fig. 1g). The SAR can reach 1083 wg^{-1} in AMF ($H = 27 \text{ kAm}^{-1}$, $f = 430 \text{ kHz}$) (Fig. 1e), which is sufficient for hyperthermia in clinical application.

Biocompatibility of $Zn_{0.5}Fe_{2.5}O_4/SiO_2$

Fig.2: The cytotoxicity of $Zn_{0.5}Fe_{2.5}O_4/SiO_2$ nanoparticles to MEF (a) and human osteosarcoma MG-63 (b) cells was assessed by CCK-8 assay. Cells without NPs were used as control groups. Concentration dependent cytotoxic effects of nanoparticles was evaluated after 24 and 48h incubation. Result are represented as mean \pm standard error of the mean.

Note: **Significant difference from control ($P < 0.01$); *** Significant difference from control ($P < 0.005$)

Cytotoxicity of $Zn_{0.5}Fe_{2.5}O_4/SiO_2$ nanoparticles to cells was measured, the cell viability of MEF and human osteosarcoma MG-63 cells was assessed by CCK-8 assay after incubation with various concentrations (50, 100, 200, 400, 600, 800, 1000 $\mu\text{g ml}^{-1}$) of NPs for 24h and 48h. As shown in Fig.2, the results demonstrated that MEF and human osteosarcoma MG-63 cells exposed to $Zn_{0.5}Fe_{2.5}O_4/SiO_2$ nanoparticles resulted in time-dependent as well as concentration-dependent cytotoxicity. There were no statistical difference ($P > 0.05$) of cell viability between treatment and control groups after 24 h incubation, even the concentration was up to 1000 $\mu\text{g ml}^{-1}$. Although viability of cells with $Zn_{0.5}Fe_{2.5}O_4/SiO_2$ is lower than control groups ($P < 0.01$) at 800 $\mu\text{g ml}^{-1}$ after 48 h

incubation, it can still reach 54.6% for MEF and 68.7% for human osteosarcoma MG-63 cells.

Biocompatibility is the precondition of achieving biomedical application with magnetic nanoparticles[33]. The cytotoxicity of NPs was generally relate to elements of nanoparticles and coating materials. Iron oxide nanoparticles are generally considered as safe, biocompatible and non-toxic materials[33]. Although biomedical applications of iron oxide nanoparticles doped with magnetically susceptible elements (e.g. MnFe_2O_4 and CoFe_2O_4) are much more restricted, because of their potential toxicity, zinc is deemed as the essential element for human being. Hence, zinc doping iron oxide nanoparticles are biocompatible and low toxicity, which has been investigated in our previous study[28]. Meanwhile, the biocompatibility is also depended on the coating materials[33]. Until now, clinical trials have been done for only two families of iron oxide nanoparticles, i. e. those coated with polysaccharides and silica[33]. The silica coating not only can decrease the cytotoxicity because of providing a stable protective layer against oxidation and reactive species[34], but also can affect the heating efficiency. Although the optimal thickness of coating is still an open question, Ansari[24] pointed out that the optimal amount of silica should be the minimum necessary to keep the nanocomposites stable in water as well as does not reduce the heat emission capability.

3.2. Cell uptake properties of $\text{Zn}_{0.5}\text{Fe}_{2.5}\text{O}_4/\text{SiO}_2$

Prussian blue staining

Fig.3. Human osteosarcoma MG-63 cells were incubated with various concentrations (a control b 6.25ug ml^{-1} , c 12.5ug ml^{-1} , d 25ug ml^{-1} , e 50ug ml^{-1} , f 100ug ml^{-1} , g 200ug ml^{-1}) of $\text{Zn}_{0.5}\text{Fe}_{2.5}\text{O}_4/\text{SiO}_2$ for 24h. The human osteosarcoma MG-63 cells were stained blue at the concentration of 6.25ug ml^{-1} and the color became deeper with the increasing of NPs concentrations.

Fig.4. Human osteosarcoma MG-63 cells were incubated with 20ug ml^{-1} $\text{Zn}_{0.5}\text{Fe}_{2.5}\text{O}_4/\text{SiO}_2$ nanoparticles for different time (a control b 0.5h, c 2h, d 4h, e 6h, f 8h, g 10h, h 12h).The cytoplasm were stained blue at 0.5h and became deeper with incubation time.

A detailed understanding of the cell uptake properties is critical when NPs were considered as magnetic hyperthermia agents. We have investigated two factors that may affect the cell uptake properties, the concentration of NPs and the incubation time. Some investigations demonstrate that the temperature is also the factor that may affect the internalization. However, the clinical application is the main purpose of magnetic hyperthermia while human being are homothermal (37°C). The temperature was considered as the inessential factor.

Firstly, human osteosarcoma MG-63 cells were incubated with various concentrations of $\text{Zn}_{0.5}\text{Fe}_{2.5}\text{O}_4/\text{SiO}_2$ nanoparticles (6.25, 12.5, 25, 50, 100, 200 ug/ml) for 24h (Figure3, a-f), the cytoplasm was stained blue at the concentration increased to 6.25ugml^{-1} , with the multiplication of the concentration, the blue color become deeper gradually. Secondly, human osteosarcoma MG-63 cells were incubated with 200ug ml^{-1} $\text{Zn}_{0.5}\text{Fe}_{2.5}\text{O}_4/\text{SiO}_2$ nanoparticles for different time (0.5, 2, 4, 6, 8, 10, 12h) (Figure4, b-h). The cytoplasm was red in the control group, when the incubation time prolong to 0.5h, the cytoplasm was stained blue. With the extension of incubation time, the blue color become deeper gradually.

Our study demonstrated that the cell uptake properties of $\text{Zn}_{0.5}\text{Fe}_{2.5}\text{O}_4/\text{SiO}_2$ are concentration dependent as well as time dependent. The NPs can internalize into the tumor cells at ultralow concentration (6.25ug ml^{-1}).

TEM

Fig.5. TEM characterization of human osteosarcoma MG-63 cells incubation with 200ug ml⁻¹ Zn_{0.5}Fe_{2.5}O₄/SiO₂ nanoparticles for 0.5h (a, b), 2h(c, d), 24h (e, f). The nanoparticles just contacted to cytomembrane and did not internalized into the cytoplasm at 0.5h. However, the NPs internalized into human osteosarcoma MG-63 cells and located in the lysosome after 2h incubation. With the extension of incubation time, the amount of NPs in lysosome increased gradually.

Transmission electron microscopy (TEM) and Prussian blue staining are the essential methods to elucidate the location and fate of NPs in cells and tissues[25], so we study the cell uptake properties further by TEM. Fig. a, b.

showed that the NPs were just contacted to the cellular membrane of the tumor cells after 0.5h incubation, there were no internalization appeared at this time point. However, internalization was observed after 2h incubation and the NPs were located in the lysosome (Fig. c, d). With the extension of incubation time, the amount of NPs were increased gradually. There were plenty of lysosomes loaded with NPs after 24h incubation (Fig. e, f). Interestingly, the NPs kept good dispersion in the lysosome at high resolution TEM image (Figure S3.), indicating they still able to heat up and damage the tumor cells and thus they are still functional as heating mediators[31]. Sadhukha[35] reported that sub-micron size aggregation induced temperature-dependent autophagy through generation of oxidative stress, micron size aggregation caused rapid membrane damage, resulting in acute cell kill. However, Cabrera[27] revealed that aggregation affect NPs magnetic characterization and heating efficiency. Hence, NPs with good dispersion rather than aggregation are benefit to keep heating efficiency inside cells. This section shows that $Zn_{0.5}Fe_{2.5}O_4/SiO_2$ can internalize into human osteosarcoma MG-63 cells, the internalization appeared about 0.5-2h after incubation, TEM image showed that the NPs were all most located in lysosome after endocytosis. It is worth to note that the NPs can still keep good distribution in the lysosome, which was benefit to heating emission efficiency when Brown relaxation was blocked.

ICP-MS

Fig.6. The amounts of iron per cell (in pg) determined by elemental analysis are reported for human osteosarcoma MG-63 cells incubation with $200\mu\text{g ml}^{-1}$ $Zn_{0.5}Fe_{2.5}O_4/SiO_2$ for different time (0.5, 2, 4, 6, 8, 10, 12h). These values provide an indication of the amount of iron internalized into the tumor cells. The iron contents were estimated by ICP-MS measurements of treated cells. All points have been acquired in triplicate.

We have also estimated the total amount of iron measured by inductively coupled plasma mass spectrometry (ICP-MS). Human osteosarcoma MG-63 cells ($1 \times 10^6 \text{ml}^{-1}$) were incubated with $200\mu\text{g ml}^{-1}$ $Zn_{0.5}Fe_{2.5}O_4/SiO_2$ nanoparticles for different time (0.5, 2, 4, 6, 8, 10, 12h). The results showed that the internalization of $Zn_{0.5}Fe_{2.5}O_4/SiO_2$ nanoparticles in human osteosarcoma MG-63 cells increased with time. However, there was not a statistically significant difference in internalization between 2h (14.23 Fe pg/cell) and 4h (18.02 pg/cell) ($P > 0.05$, $\alpha = 0.05$), but there was a statistically significant difference among other time points ($P < 0.05$, $\alpha = 0.05$). Interestingly, the results are consistent with Prussian blue staining, which also showed time-dependent internalization. However, the TEM showed that NPs just contacted with the cytomembrane and did not internalize into human osteosarcoma MG-63 cells at 0.5h, we speculate that the reason may because of the extracellular NPs are difficult to rinse completely.

3.3. Relationship of Cell Uptake and Magnetic Hyperthermia Efficiency

Magnetic hyperthermia represents a novel therapeutic method to cancer treatment and is based on the mechanism that cancer cells are more sensitive than healthy cells to temperature higher than 43°C [36]. Compare to various approaches proposed to raise the body temperature[37], magnetic hyperthermia can offer several advantages[31], One of which is magnetic hyperthermia can induce tumor cells to death without damage healthy

tissue. The main reason is that NPs can internalize into tumor cells through endocytosis and realize “inside-out”
hyperthermia. Prior studies demonstrated that iron oxide nanoparticles (IO-MNPs) exposed to an AMF respond by
locally releasing heat and/or mechanically rotating[38]. EGFR targeted magnetic nanoparticles in AMF were
shown to significantly induce cancer cells to death without a perceptible macroscopic temperature rise[17]. Zhang
et al[39]covalently conjugated SPIONs with antibodies targeting the lysosomal protein marker
LAMP1(LAMP-SPIONs) and induced tumor cells to apoptosis through tear of lysosome membrane in a unique
dynamic magnetic field (DMF). Maribella et al[38]synthesized iron oxide magnetic nanoparticles (MNPs) targeted
to the epidermal growth factor receptor(EGFR) exposed to AMF that can selectively induce lysosomal membrane
permeabilization (LMP) in cancer cells overexpressing the EGFR, despite the SAR of these NPs is low
($175\pm 18\text{Wg}^{-1}$, $f=233\text{kHz}$, $H 40\text{kAm}^{-1}$). Thus, we further investigate the relationship between the internalization
and hyperthermia performance, then try to figure out which mechanism acts as the main role that induces tumor
cells to death.

Such $\text{Zn}_{0.5}\text{Fe}_{2.5}\text{O}_4/\text{SiO}_2$ nanoparticles can be useful for magnetic hyperthermia since its good biocompatibility and
high SAR. Cell apoptosis was examined by flow cytometry based on Annexin-V-FITC/PI Apoptotic Assay, as
shown in Table.1-2 and Fig.7a. Human osteosarcoma MG-63 cells without NPs and human osteosarcoma MG-63
cells with NPs but without AMF were used as negative and positive control groups respectively. Human
osteosarcoma MG-63 cells with NPs and exposed to AMF (430 kHz , 11kA m^{-1}) were the treatment groups. Each
treatment groups contain “No rinse” and “Rinse” group which was determined by the incubation time (0.5h, 2h, 6h,
10h). Table 1-2 show that the percentage of the apoptotic cells of “No rinse” groups are obviously higher than
“Rinse” group at each time points. In “No rinse” groups, the cell killing efficiency do not increase with the
incubation time, the 0.5h and 2h groups are higher than 6h and 10h groups. The percentage of apoptotic cells of 2h
is 61% in total, while that is 41.3% in 10h group. However, in “Rinse” group, although the cell killing efficiency
do not show any pattern with the incubation time, the 10h Rinse group show the highest cell killing efficiency. It is
worth to mention that the late apoptosis is the main pathway of MHT in all groups. The results of flow cytometry
are consistent with fluorescence imaging (Fig.7b). The microstructure of human osteosarcoma MG-63 cells after
MHT by $\text{Zn}_{0.5}\text{Fe}_{2.5}\text{O}_4/\text{SiO}_2$ (Fig.7c) showed that the NPs came out from the ruptured lysosome and the
cytomembrane were also damaged, but the chromatin appeared edge aggregation while the nuclear membranes
were still intact. These features are the characterization of apoptosis. Our study demonstrated that
$\text{Zn}_{0.5}\text{Fe}_{2.5}\text{O}_4/\text{SiO}_2$ nanoparticles can be the candidate of magnetic hyperthermia agents with a low concentration.
However, the hyperthermia efficiency did not increase with the increasing of NPs in the cells. The 2h “No rinse”
group show the highest cell apoptosis rate while the 10h “No rinse” group was the lowest group. The decrease of
hyperthermia efficiency may cause by the block of Brownian relaxation. 35 With the increasing of internalized NPs, the block effect of Brownian relaxation enhanced.  It is worth to note that
the cell apoptosis rate of 2h “No rinse” group is also higher than 0.5h”No rinse” group. On basis of the TEM
imaging, we have known that the NPs are just contact with the cytomembrane and do not internalize into tumor
cells at 0.5h incubation. Consequently, the best policy to enhance hyperthermia efficiency of NPs is not saturated
but appropriate amount NPs in cells. Interestingly, although the extracellular NPs were rinsed, tumor cells could
still be killed by NPs in the lysosomes. However, the cell kill efficiency of the “Rinse” group is obviously lower
than the “No rinse” group, and we still can monitor the macroscopic temperature rise (below 40°C). Hence, we
speculate that the local temperature rise rather than NPs mechanical rotation  is the main role that induces tumor
cells to death.

Table 1. Apoptotic Assay of human osteosarcoma MG-63, induced by MHT of Zn_{0.5}Fe_{2.5}O₄@SiO₂ (No rinse group)

	Control	NPs	0.5h	2h	6h	10h
Q1	0.058%	0.015%	1.89%	11.3%	5.26%	6.57%
Q2	0.097%	0.255%	43.3%	57.9%	43.7%	34.5%
Q3	0.335%	3.5%	17.7%	8.13%	9.54%	6.83%
Q4	99.5%	97.7%	37.1%	22.6%	41.5%	50.5%

Table 2. Apoptotic Assay of human osteosarcoma MG-63, induced by MHT of Zn_{0.5}Fe_{2.5}O₄@SiO₂ (Rinse group)

	Control	NPs	0.5h	2h	6h	10h
Q1	0.058%	0.015%	0.65%	0.339%	0.259%	0.299%
Q2	0.097%	0.255%	5.25%	5.67%	5.49%	7.56%
Q3	0.335%	3.5%	2.79%	1.61%	2.18%	3.44%
Q4	99.5%	97.7%	91.3%	92.4%	92.1%	88.7%

(a)

(b)

(c)

Fig.7 a Apoptotic Assay of human osteosarcoma MG-63 cells incubated with $Zn_{0.5}Fe_{2.5}O_4/SiO_2$ for different time (0.5h, 2h, 6h, 10h) after exposing to AMF (f 430 kHz, H 11 kAm⁻¹). Human osteosarcoma MG-63 cells without NPs or AMF were used as a control group. Human osteosarcoma MG-63 cells with NPs but without AMF were used as another control group (NPs group). b The fluorescence imaging of human osteosarcoma MG-63 cells after MHT. c TEM imaging of human osteosarcoma MG-63 cells incubated with $Zn_{0.5}Fe_{2.5}O_4/SiO_2$ for 2h after magnetic hyperthermia.

4. Conclusion

We have successfully synthesized monodispersed $Zn_{0.5}Fe_{2.5}O_4$ nanoparticles of 22 nm by one-pot approach and coated with silica as magnetic hyperthermia agents. The $Zn_{0.5}Fe_{2.5}O_4/SiO_2$ nanoparticles have been characterized as superparamagnetic materials with high SAR and Ms. The results have showed that their excellent colloidal stability and low cytotoxicity, which can be considered as magnetic fluid hyperthermia candidate. The cell uptake properties investigation have demonstrated that such NPs can internalized into human osteosarcoma MG-63 cells, the internalization appeared between 0.5 to 2h and the NPs mostly located in lysosome after endocytosis. Furthermore, the hyperthermia performance is related to amount of internalized NPs, but the best amount is appropriate rather than saturated, which is worth to further study. The NPs can still induce tumor cell to death when extracellular NPs were rinsed. However, the local temperature rise rather than NPs mechanical rotation is considered as the main factor that induce tumor cell to death. These results promote us to connect these NPs with

special organella targeted agents that will enhance the temperature and mechanical mechanism both in the future.

Ethics. This article does not present research with ethical considerations.

Data accessibility. The supplementary information and original experiment data of this study are available within the Dryad Digital Repository: <https://doi.org/10.5061/dryad.sr0g5s1>.

Reviewer URL: <https://datadryad.org/review?doi=doi:10.5061/dryad.sr0g5s1>

Authors' Contributions. Keya Mao and Peifu Tang designed the study. Runsheng Wang and Yihao Liu accomplished whole cell experiment. Jianheng Liu, Rui Zhong and Qingzu Liu collected and analysed the data. Xiang Yu, Li Zhang and Chenhui Lv were responsible for synthesis and coating of magnetite nanoparticles. Runsheng Wang interpreted the results and wrote the manuscript. All author gave final approval for publication.

Competing interests. The authors declare no competing interests.

Funding. This work was supported by National Natural Science Foundation of China (Grant No.51772328, 81702121) and Natural Science Foundation of Guangxi Zhuang Autonomous Region (Grant No.2018JJB140367)

Acknowledgements. We are very grateful to Professor Shuli He and the staff of department of physics of Capital Normal University for guidance in the synthesis of nanoparticles. We thank Lin Chen, Shulong Yuan and Jin Li of Translational Medical Center of PLA General Hospital for careful laboratory assistance.

References

1. L. G. Wence Ding. 2013 Immobilized transferrin Fe₃O₄@SiO₂ nanoparticle with high doxorubicin loading for dual-targeted tumor drug delivery. *Int J Nanomedicine*. **8**, 4631-4639. (doi:10.2147/ijn.s51745).
2. D.-L. He. 2017 Ten Things You Might Not Know about Iron Oxide Nanoparticles. *Radiology*. **284(3)**, 616-629. (doi:10.1148/radiol.2017162759).
3. H. Chen, J. Shen, E. Choy, F. J. Hornicek, Z. Duan. 2016 Targeting protein kinases to reverse multidrug resistance in sarcoma. *Cancer Treat Rev*. **43**, 8-18. (doi:10.1016/j.ctrv.2015.11.011).
4. J. J. Lee, K. J. Jeong, M. Hashimoto, A. H. Kwon, A. Rwei, S. A. Shankarappa, J. H. Tsui, D. S. Kohane. 2014 Synthetic ligand-coated magnetic nanoparticles for microfluidic bacterial separation from blood. *Nano Lett*. **14(1)**, 1-5. (doi:10.1021/nl3047305).
5. Y. A. Chenyan Yuan, Jia Zhang, Hongbo Li, Hao Zhang, Ling Wang, Dongsheng Zhang. 2014 Magnetic nanoparticles for targeted therapeutic gene delivery and magnetic-inducing heating on hepatoma. *Nanotechnology*. **25(34)**, 345101. (doi:10.1088/0957-4484/25/34/345101).
6. F. Wang, Y. Yang, Y. Ling, J. Liu, X. Cai, X. Zhou, X. Tang, B. Liang, Y. Chen, H. Chen, D. Chen, C. Li, Z. Wang, B. Hu, Y. Zheng. 2017 Injectable and thermally contractible hydroxypropyl methyl cellulose/Fe₃O₄ for magnetic hyperthermia ablation of tumors. *Biomaterials*. **128**, 84-93. (doi:10.1016/j.biomaterials.2017.03.004).
7. B. H. P. Wust, G. Sreenivasa, B. Rau, J. Gellermann, H. Riess, R. Felix, P. Schlag. 2002 Hyperthermia in combined treatment of cancer. *The Lancet Oncology*. **3(8)**, 487-497. (doi:10.1016/s1470-2045(02)00818-5).
8. R. T. Gordon, J. R. Hines, D. Gordon. 1979 Intracellular hyperthermia. A biophysical approach to cancer treatment via intracellular temperature and biophysical alterations. *Medical Hypotheses*. **5(1)**, 83-102. (doi:10.1016/0306-9877(79)90063-x).
9. N. R. Datta, S. Krishnan, D. E. Speiser, E. Neufeld, N. Kuster, S. Bodis, H. Hofmann. 2016 Magnetic nanoparticle-induced hyperthermia with appropriate payloads: Paul Ehrlich's "magic (nano)bullet" for cancer theranostics? *Cancer Treat Rev*. **50**, 217-227. (doi:10.1016/j.ctrv.2016.09.016).
10. B. Sanz, M. P. Calatayud, T. E. Torres, M. L. Fanarraga, M. R. Ibarra, G. F. Goya. 2017 Magnetic

hyperthermia enhances cell toxicity with respect to exogenous heating. *Biomaterials*. **114**, 62-70.
(doi:10.1016/j.biomaterials.2016.11.008).
11. B. P. Shah, N. Pasquale, G. De, T. Tan, J. Ma, K. B. Lee. 2014 Core-shell nanoparticle-based peptide
therapeutics and combined hyperthermia for enhanced cancer cell apoptosis. *Acs Nano*. **8(9)**, 9379-9387.
(doi:10.1021/nn503431x).
12. G. Baldi, C. Ravagli, F. Mazzantini, G. Loudos, J. Adan, M. Masa, D. Psimadas, E. A. Fragogeorgi, E.
Locatelli, C. Innocenti, C. Sangregorio, M. C. Franchini. 2014 In vivo anticancer evaluation of the
hyperthermic efficacy of anti-human epidermal growth factor receptor-targeted PEG-based nanocarrier
containing magnetic nanoparticles. *Int J Nanomedicine*. **9**, 3037-3056. (doi:10.2147/ijn.s61273).
13. A. Espinosa, R. Di Corato, J. Kolosnjaj-Tabi, P. Flaud, T. Pellegrino, C. Wilhelm. 2016 Duality of Iron Oxide
Nanoparticles in Cancer Therapy: Amplification of Heating Efficiency by Magnetic Hyperthermia and
Photothermal Bimodal Treatment. *Acs Nano*. **10(2)**, 2436-2446. (doi:10.1021/acsnano.5b07249).
14. A. Espinosa, M. Bugnet, G. Radtke, S. Neveu, G. A. Botton, C. Wilhelm, A. Abou-Hassan. 2015 Can
magneto-plasmonic nanohybrids efficiently combine photothermia with magnetic hyperthermia? *Nanoscale*.
**7(45)**, 18872-18877. (doi:10.1039/c5nr06168g).
15. S. M. Min Ka, Yu F, Yang M, David Ae, Yang Vc, Rosania Gr. 2013 Pulsed Magnetic Field Improves the
Transport of Iron Oxide Nanoparticles through Cell Barriers. *Acs Nano*. **7(3)**, 2161-2171.
(doi:org/10.1021/nn3057565).
16. E. Zhang, M. F. Kircher, M. Koch, L. Eliasson, S. N. Goldberg, E. Renstrom. 2014 Dynamic Magnetic Fields
Remote-Control Apoptosis via Nanoparticle Rotation. *Acs Nano*. **8(4)**, 3192-3201. (doi:10.1021/nn406302j).
17. A. C. B. Mar Creixell, Madeline Torres-Lugo, Carlos Rinaldi. 2011 EGFR-Targeted Magnetic Nanoparticle
Heaters Kill Cancer Cells Without A Sensible Temperature Rise. *Acs Nano*. **5(9)**, ACS Nano.
(doi:org/10.1021/nn201822b).
18. B. Sanz, M. Pilar Calatayud, E. De Biasi, E. Lima, Jr., M. Vasquez Mansilla, R. D. Zysler, M. Ricardo
Ibarra, G. F. Goya. 2016 In Silico before In Vivo: how to Predict the Heating Efficiency of Magnetic
Nanoparticles within the Intracellular Space. *Sci Rep*. **6**. (doi:10.1038/srep38733).
19. D. Soukup, S. Moise, E. Cespedes, J. Dobson, N. D. Telling. 2015 In situ measurement of magnetization
relaxation of internalized nanoparticles in live cells. *Acs Nano*. **9(1)**, 231-240. (doi:10.1021/nn503888j).
20. R. Di Corato, A. Espinosa, L. Lartigue, M. Tharaud, S. Chat, T. Pellegrino, C. Menager, F. Gazeau, C.
Wilhelm. 2014 Magnetic hyperthermia efficiency in the cellular environment for different nanoparticle designs.
*Biomaterials*. **35(24)**, 6400-6411. (doi:10.1016/j.biomaterials.2014.04.036).
21. S. Dutz, M. Kettering, I. Hilger, R. Mueller, M. Zeisberger. 2011 Magnetic multicore nanoparticles for
hyperthermia-influence of particle immobilization in tumour tissue on magnetic properties. *Nanotechnology*.
**22(26)**. (doi:10.1088/0957-4484/22/26/265102).
22. R. Hergt, S. Dutz. 2007 Magnetic particle hyperthermia—biophysical limitations of a visionary tumour
therapy. *Journal of Magnetism and Magnetic Materials*. **311(1)**, 187-192. (doi:10.1016/j.jmmm.2006.10.1156).
23. L. Ye, K. T. Yong, L. Liu, I. Roy, R. Hu, J. Zhu, H. Cai, W. C. Law, J. Liu, K. Wang, J. Liu, Y. Liu, Y. Hu, X.
Zhang, M. T. Swihart, P. N. Prasad. 2012 A pilot study in non-human primates shows no adverse response to
intravenous injection of quantum dots. *Nat Nanotechnol*. **7(7)**, 453-458. (doi:10.1038/nnano.2012.74).
24. L. Ansari, B. Malaekheh-Nikouei. 2016 Magnetic silica nanocomposites for magnetic hyperthermia
applications. *Int J Hyperthermia*, 1-34. (doi:10.1080/02656736.2016.1243736).
25. T. A. Bogart Lk, Cesbron Y, Murray P, Lévy R. 2012 Photothermal Microscopy of the Core of
Dextran-Coated Iron Oxide Nanoparticles During Cell Uptake. *Acs Nano*. **6(7)**, 5961-5971.
(doi:org/10.1021/nn300868z).

26. L. E. J. Silva Ah, Mansilla Mv, Zysler Rd, Troiani H, Piscioti Mlm, Locatelli C, Benech Jc, Oddone N, Zoldan Vc, Winter E, Pasa Aa, Creczynski-Pasa Tb. 2016 Superparamagnetic iron-oxide nanoparticles mPEG350- and mPEG2000-coated: cell uptake and biocompatibility evaluation. *Nanomedicine*. **12(4)**, 909-919. (doi:10.1016/j.nano.2015.12.371).
 27. D. Cabrera, A. Coene, J. Leliaert, E. J. Artes-Ibanez, L. Dupre, N. D. Telling, F. J. Teran. 2018 Dynamical Magnetic Response of Iron Oxide Nanoparticles Inside Live Cells. *Acs Nano*. **12(3)**, 2741-2752. (doi:10.1021/acsnano.7b08995).
 28. S. He, H. Zhang, Y. Liu, F. Sun, X. Yu, X. Li, L. Zhang, L. Wang, K. Mao, G. Wang, Y. Lin, Z. Han, R. Sabirianov, H. Zeng. 2018 Maximizing Specific Loss Power for Magnetic Hyperthermia by Hard-Soft Mixed Ferrites. *Small*, e1800135. (doi:10.1002/sml.201800135).
 29. C. Blanco-Andujar, A. Walter, G. Cotin, C. Bordeianu, D. Mertz, D. Felder-Flesch, S. Begin-Colin. 2016 Design of iron oxide-based nanoparticles for MRI and magnetic hyperthermia. *Nanomedicine (Lond)*. **11(14)**, 1889-1910. (doi:10.2217/nmm-2016-5001).
 30. S. H. Noh, W. Na, J. T. Jang, J. H. Lee, E. J. Lee, S. H. Moon, Y. Lim, J. S. Shin, J. Cheon. 2012 Nanoscale magnetism control via surface and exchange anisotropy for optimized ferrimagnetic hysteresis. *Nano Lett*. **12(7)**, 3716-3721. (doi:10.1021/nl301499u).
 31. D. C. R. Guardia P, Lartigue L, Wilhelm C, Espinosa a, Garcia-Hernandez M, Gazeau F, Manna L, Pellegrino T. 2012 Water-Soluble Iron Oxide Nanocubes with High Values of Specific Absorption Rate for Cancer Cell Hyperthermia Treatment. *Acs Nano*. **6(4)**, 3080-3091. (doi:10.1021/nn2048137).
 32. T. Zargar, A. Kermanpur. 2017 Effects of hydrothermal process parameters on the physical, magnetic and thermal properties of Zn_{0.3}Fe_{2.7}O₄ nanoparticles for magnetic hyperthermia applications. *Ceramics International*. **43(7)**, 5794-5804. (doi:10.1016/j.ceramint.2017.01.127).
 33. H. Arami, A. Khandhar, D. Liggitt, K. M. Krishnan. 2015 In vivo delivery, pharmacokinetics, biodistribution and toxicity of iron oxide nanoparticles. *Chem Soc Rev*. **44(23)**, 8576-8607. (doi:10.1039/c5cs00541h).
 34. D. Ling, T. Hyeon. 2013 Chemical design of biocompatible iron oxide nanoparticles for medical applications. *Small*. **9(9-10)**, 1450-1466. (doi:10.1002/sml.201202111).
 35. T. Sadhukha, T. S. Wiedmann, J. Panyam. 2014 Enhancing therapeutic efficacy through designed aggregation of nanoparticles. *Biomaterials*. **35(27)**, 7860-7869. (doi:10.1016/j.biomaterials.2014.05.085).
 36. R. Hergt, S. Dutz, R. Müller, M. Zeisberger. 2006 Magnetic particle hyperthermia: nanoparticle magnetism and materials development for cancer therapy. *Journal of Physics: Condensed Matter*. **18(38)**, S2919-S2934. (doi:10.1088/0953-8984/18/38/s26).
 37. L. Wang, P. Zhang, J. Shi, Y. Hao, D. Meng, Y. Zhao, Y. Yanyan, D. Li, J. Chang, Z. Zhang. 2015 Radiofrequency-triggered tumor-targeting delivery system for theranostics application. *ACS Appl Mater Interfaces*. **7(10)**, 5736-5747. (doi:10.1021/am507898z).
 38. I. M.-B. Maribella Domenech, Madeline Torres-Lugo, Carlos Rinaldi. 2013 Lysosomal Membrane Permeabilization by Targeted Magnetic Nanoparticles in Alternating Magnetic Fields. *Acs Nano*. **7(6)**, 5091-5101. (doi:10.1021/nn4007048).
 39. K. M. Zhang E, Koch M, Eliasson L, Goldberg Sn, Renström E. 2014 Dynamic Magnetic Fields Remote-Control Apoptosis via Nanoparticle Rotation. *Acs Nano*. **8(4)**, 3192-3201. (doi:10.1021/nn406302j).

176x197mm (161 x 161 DPI)

176x62mm (300 x 300 DPI)

176x88mm (226 x 226 DPI)

176x88mm (225 x 225 DPI)

82x64mm (300 x 300 DPI)

176x87mm (300 x 300 DPI)

Table 1. Apoptotic Assay of MG-63, induced by MHT of Zn_{0.5}Fe_{2.5}O₄@SiO₂ (No rinse group)

	Control	NPs	0.5h	2h	6h	10h
Q1	0.058%	0.015%	1.89%	11.3%	5.26%	6.57%
Q2	0.097%	0.255%	43.3%	57.9%	43.7%	34.5%
Q3	0.335%	3.5%	17.7%	8.13%	9.54%	6.83%
Q4	99.5%	97.7%	37.1%	22.6%	41.5%	50.5%

Table 2. Apoptotic Assay of MG-63, induced by MHT of Zn_{0.5}Fe_{2.5}O₄@SiO₂ (Rinse group)

	Control	NPs	0.5h	2h	6h	10h
Q1	0.058%	0.015%	0.65%	0.339%	0.259%	0.299%
Q2	0.097%	0.255%	5.25%	5.67%	5.49%	7.56%
Q3	0.335%	3.5%	2.79%	1.61%	2.18%	3.44%
Q4	99.5%	97.7%	91.3%	92.4%	92.1%	88.7%

Appendix B

Dear Dr. Andrew Dunn,

On behalf of my co-authors, we thank you very much for giving us an opportunity to revise our manuscript (ID RSOS-191139.R1). We appreciate you and reviewers very much for the positive comments and advice. These comments are all valuable and very helpful for revising and improving our manuscript.

According to the editors' and reviewers' recommendations, we have revised our manuscript carefully under the revision status and completed submission of the revised manuscript. The following is a detailed list of response to all comments and suggestions.

Once again, we deeply appreciate your consideration of our manuscript, and we look forward to your reply. If you have any queries, please do not hesitate to contact me at the address below.

Thank you and best regards,

Sincerely yours,

Keya Mao

Medical School of Chinese PLA, Beijing 100853, Corresponding author: Keya Mao

E-mail: jianhengliu@126.com.

Response to Reviewers

To Reviewers:

Firstly, thank you to our affirmation. Secondly, we thank you very much for your comments and suggestions again. Your comments and suggestions are constructive for revising and improving our manuscript. According to your suggestions, we have revised our manuscript. Please review.

Reviewer 1: This is an interesting work with a lot of experimental validation. My only concerns are that some interpretations require more solid justifications. See attached file (manuscript pdf file with comments) for more details.

For example,

1. what is the beneficiary role of Zn substitute to this specific MNPs

Answer: Firstly, the saturation magnetization (M_s) of zinc ferrite is sensitive to Zn content, with $ZnFe_2O_4$ being antiferromagnetic Zn^{2+} ions occupy the A-site of the spinel lattice only. The M_s and anisotropy can be tuned by varying the Zn: Fe ratio, as we have investigated in our previous study (*doi:10.1002/sml.201800135*), the M_s can be increased by doping zinc in ferrite properly. We found that the M_s of zinc doping iron oxide nanoparticles were higher than Fe_3O_4 nanoparticles. Therefore, the heating efficiency of zinc doping nanoparticles is higher than Fe_3O_4 nanoparticles in our experiment. Secondly, zinc doping iron oxide nanoparticles are biocompatible, which are suitable for biomedical application.

2. Why 2h incubation is the optimum time for internalization?

Answer: The cell uptake properties of $Zn_{0.5}Fe_{2.5}O_4/SiO_2$ were concentration dependent as well as time dependent. As shown in Fig.6, the amounts of iron internalized into the tumor cells were increased with time, but the 2h incubation was not the maximum amounts internalization. However, the 2h “No rinse” group showed the highest cell apoptosis rate. The internalized NPs of 6h and 10h are more than the 0.5h and 2h. Therefore, there are more NPs hindered in 6h and 10h groups than 0.5h and 2h groups, then the heating efficiency of 6h and 10h is weaker than 0.5h and 2h. Meanwhile, the cell apoptosis rate of 2h “No rinse” group was also higher than 0.5h “No rinse” group. Hence, the best policy to enhance hyperthermia efficiency of NPs is not saturated but appropriate amount NPs in cells, but the mechanism need to be studied further.

3. Hyperthermia interpretation requires more careful comments, than the general one that Brownian relaxation is hindered.

Answer: The Brownian relaxation is related to the viscosity of surrounding liquid and the hydrodynamic diameter of the NPs. As shown in Fig.5, the NPs internalized into the tumor cells and located in lysosomes. However, the viscosity of lysosomes content is thicker than deionized water, but thinner than agar, which is generally used in the experiment of hindering Brownian relaxation. Therefore, we speculate that the block effect of Brownian relaxation is incomplete. Simultaneously, the NPs come out from the ruptured tumor cells after intracellular hyperthermia and the Brownian relaxation will restore (*10.1021/nm503888j*). Thank you for this suggestion and we will revise the interpretation of hyperthermia.

4. Why rinse and no rinse samples exhibit distinctly different features?

Answer: The distinctly different features may because that the location of main heat mediators is different. In no rinse samples, the extracellular and intracellular NPs are all the heat mediator, but the extracellular NPs act as the main role because the extracellular NPs are much more than intracellular NPs. As shown in Fig.6, the internalized NPs of 6h and 10h are more than 0.5h and 2h. For the block of Brownian relaxation, the heating efficiency of 6h and 10h no rinse samples is weaker than 0.5h and 2h. So, the apoptosis rate of 0.5h and 2h no rinse samples are higher than 6h and 10h. However, in rinse samples, the intracellular NPs are the only heat mediator. Although the Brownian relaxation is hindered for all rinse samples, they still can undergo Néel relaxation (*doi:10.1021/nm503888j*). As mentioned above, the internalized NPs of 10h rinse samples are more than the other rinse samples, so the heating efficiency of 10h rinse samples is higher than the other rinse samples. Therefore, the apoptosis rate of 10h is the highest in all rinse samples.

5. Finally, I have located a lot of grammar mistakes. The manuscript should undergo a careful English grammar and syntax checking by a language expert.

Answer: Thank you for your constructive suggestions. We check the whole

manuscript and revise it carefully according to your comments.

Reviewer 2: I have no doubt that the paper is interesting. But I am also convinced that it needs a lot of edition, as the language is unclear in many places. A thorough revision of English is first of all recommended. Other observations follow:

1. Page 3, line 18: What is the meaning of DMF abbreviation?

Answer: DMF means dynamic magnetic fields (*doi:10.1021/nn406302j*).

2. Page 3, line 55: although this is provided later in the paper, some information regarding toxicity of Zn would be of interest Why silica?

Answer: Although crystalline silica has been suspected human carcinogen and involved in the pathogenesis of silicosis, amorphous silica is approved by US FDA as food additive (*doi:10.1080/02656736.2016.1243736*). Up to now, there are only two families of IONPs used in clinical trials, which are coated with polysaccharides and silica (*doi:10.1039/c5cs00541h*). In our study, silica was added to the solution to provide a surface coating material as its good biocompatibility. Consequently, the toxicity of Zn would be of interest rather than silica.

3. P6 L4: country of MSI company?

Answer: USA

4. P 6, L 24: is this field-frequency combination within admitted limits?

Answer: We found that it was very difficult to maintain the hyperthermia temperature at the point of 43°C, the hyperthermia temperature needed to be adjust in the whole process. We tried to adjust the temperature by change the magnitude and maintain the temperature between 43 to 45°C. For the safety of patients, the product of magnitude and frequency of AMF should be less than $5 \times 10^9 \text{ Am}^{-1}\text{s}^{-1}$ (*doi:10.1088/0953-8984/18/38/s26*). Most of the magnitude in our study is about 10 to 11kAm⁻¹, 12-14 kAm⁻¹ is seldom used only when the temperature decreased sharply. At the meantime, there are also some literatures suspect the safety limit of AMF, the magnetic field in their cell hyperthermia experiment have exceeded the safety limit (*doi:10.1021/nl301499u*). Therefore, the safety limit may need to be verified further in the future.

5. P8 L3: 6.0+-1.8

Answer: It is our mistake and we have revised the data.

6. P8 L28: the value of Ms is not large enough for justifying the huge SAR

Answer: The SAR is not only related to Ms, but also related to magnetic anisotropy constant. In our study, we try to acquire optimum SAR by modulating the magnetic anisotropy. However, the Mn and Co ions have to be added to nanoparticles if we want to obtain the highest SAR (*doi:10.1038/NNANO.2011.95*). Some literatures have demonstrated that the Mn and Co ions are considered as toxic, so we try to avoid adding Mn and Co ions into nanoparticles for the purpose of biomedical application.

Therefore, the value of M_s in our study can reach such high SAR is due to modulate the magnetic anisotropy. If we add Mn and Co ions into the nanoparticles, the SAR even can reach 3417 Wg^{-1} , which has been demonstrated in our previous study ([doi:10.1002/sml.201800135](https://doi.org/10.1002/sml.201800135)). The aim of this paper is to emphasize the biomedical application, so we consider the biocompatibility is much more important. That is why we do not choose the nanoparticles with the highest SAR as our study object. In addition, the saturation magnetization (M_s) of zinc ferrite is sensitive to Zn content, with ZnFe_2O_4 being antiferromagnetic Zn^{2+} ions occupy the A-site of the spinel lattice only. The M_s and anisotropy can be tuned by varying the Zn: Fe ratio, the SAR can only be enhanced by appropriate ration.

7. The results in Fig 2 seem to indicate cytotoxicity

Answer: As shown in Fig.2, the nanoparticles indicate cytotoxicity when the concentration increases to $800 \mu\text{g ml}^{-1}$ after 48h incubation. In fact, the cytotoxicity is related to the nanoparticles concentration and incubation time. Although previous literatures have verified the good biocompatibility of their nanoparticles ([doi:10.1016/j.nano.2013.11.01](https://doi.org/10.1016/j.nano.2013.11.01); [doi:10.1021/nn2048137](https://doi.org/10.1021/nn2048137); [doi:10.1016/j.biomaterials.2014.07.019](https://doi.org/10.1016/j.biomaterials.2014.07.019)), such good biocompatibility are conditional, the concentration and incubation time should be controlled. The nanoparticles may indicate toxicity with higher concentration and longer incubation time. According to the results of cytotoxicity, we chose $400 \mu\text{g ml}^{-1}$ nanoparticles to carry on the intracellular hyperthermia experiment and try to avoid the affection of cytotoxicity to tumor cells. However, the safety concentration of nanoparticles and the long term toxicity in vivo need to be studied further.

8. Fig 4: are the particle INSIDE or AROUND the cells?

Answer: The nanoparticles are inside the cells. Fig.4 shows the Prussian blue staining of internalized nanoparticles in the tumor cells. The cell nucleus are stained red by eosin, while the cytoplasm are stained blue by Prussian blue, the border of the cell nucleus is clear. If the nanoparticles are around the cells, the border of the nucleus will be unclear.

9. P12 L56: not so novel technique. Also: this a very interesting part of the paper, but very unclear because of the language. Special care is suggested here

Answer: Thank you for your constructive suggestions. As you said, this part is important for it is the summary of the former investigation. We have revised it carefully. Please review.